# Domain Adaptive Imitation Learning
# with Visual Observation

**Sungho Choi**[1]        **Seungyul Han**[2][*]        **Woojun Kim**[3]
**Jongseong Chae**[1]        **Whiyoung Jung**[4]        **Youngchul Sung**[1]

[1]KAIST        [2]UNIST        [3]Carnegie Mellon University        [4]LG AI Research
{sungho.choi,jongseong.chae,ycsung}@kaist.ac.kr,
syhan@unist.ac.kr, woojunk@andrew.cmu.edu, whiyoung.jung@lgresearch.ai

## Abstract

In this paper, we consider domain-adaptive imitation learning with visual observation, where an agent in a target domain learns to perform a task by observing expert demonstrations in a source domain. Domain adaptive imitation learning arises in practical scenarios where a robot, receiving visual sensory data, needs to mimic movements by visually observing other robots from different angles or observing robots of different shapes. To overcome the domain shift in cross-domain imitation learning with visual observation, we propose a novel framework for extracting domain-independent behavioral features from input observations that can be used to train the learner, based on dual feature extraction and image reconstruction. Empirical results demonstrate that our approach outperforms previous algorithms for imitation learning from visual observation with domain shift.

## 1   Introduction

Imitation learning (IL) is a framework where an agent learns behavior by mimicking an expert's demonstration without access to true rewards. The mainstream IL methods include Behavior Cloning (BC), which trains the learner with supervised learning [3, 31, 33, 39], Inverse Reinforcement Learning (IRL), which tries to find the reward function [1, 29, 49], and Adversarial Imitation Learning (AIL), which trains the learner to match its state-action visitation distribution to that of the expert via adversarial learning [10, 12, 18, 40, 46]. IL provides an effective alternative to solving reinforcement learning (RL) problems because it circumvents the difficulty in the careful design of a reward function for intended behavior [30, 47]. Despite the effectiveness, conventional IL assumes that the expert and the learner lie in the same domain, but this is a limited assumption in the real world. For example, a self-driving car should learn real-world driving skills from demonstrations in a driving simulator. In such a case, the expert and the learner do not lie in the same domain.

In this paper, we consider the challenge of domain shift in IL, i.e., an agent in a target domain learns to perform a task based on an expert's demonstration in a source domain. In particular, we focus on the case where the expert demonstrations are provided in a visual form. In many practical cases, obtaining expert demonstrations as an explicit sequence of states (such as positions and velocities) and actions (such as movement angles) can be challenging. Instead, the expert's demonstrations are often provided as image sequences. One example can be a robot mimicking a movement by visually observing human behavior. However, learning from visual observations poses a major challenge in IL since images are high-dimensional and often include only partial information about the expert's behavior or information unrelated to the expert's behavior. Even minor changes between the source and target domains can vastly affect the agent's policy and lead to unstable learning. In recent years,

---

[*]Corresponding Author

37th Conference on Neural Information Processing Systems (NeurIPS 2023).

several works aimed to address the challenge of domain shift in IL. Some methods show impressive performance by training a model to learn a mapping between domains [22, 32], requiring proxy tasks (i.e., additional tasks to assist in learning the target task) to provide expert demonstrations in both source and target domains. However, this approach assumes expert demonstrations for similar tasks in the target domain, which may not be practical in many realistic domain transfers where expert demonstrations in the target domain are unavailable. Other methods generate metric-based rewards that minimize the distance between the expert and the learner trajectories [9, 27], assuming the expert's state is directly available rather than requiring image observations. Our focus is on solving domain shifts with visual observations without the use of proxy tasks or direct access to expert's states. Our work is closely related to the methods that learn a model to extract domain-invariant features [6, 37]. However, removing domain information was not satisfactory in situations with a large domain gap. In contrast, we propose a novel learning method to train the learner in domain-adaptive IL with visual observation, achieving significantly improved performance in IL tasks with domain shift.

## 2 Related Work

**IL with domain shift**   IL with domain shift is a challenging problem where the learner should learn from an expert in a source domain and apply the learned policy in a target domain. In recent years, there has been significant progress in addressing this problem in machine learning, RL, and robotics [6, 7, 9, 11, 14, 22, 27, 28, 32, 34–37, 45]. For example, Fickinger et al. [9] proposed utilizing Gromov-Wasserstein distance to generate reward minimizing the difference between expert and learner policies. Franzmeyer et al. [11] learned a mapping between two domains, where expert embeddings are reduced by selecting task-relevant state elements based on mutual information criterion. These methods assume to have access expert's states; in contrast, our method assumes expert demonstrations are only available as high-dimensional image sequences with partial information about the expert. Kim et al. [22] and Raychaudhuri et al. [32] trained a model that can map between two domains by using expert demonstrations for proxy tasks on both domains. In comparison, our method does not rely on expert demonstrations of proxy tasks. Sermanet et al. [34] employed a model that learns representations from visual demonstrations of different viewpoints using a time-contrastive approach. This approach mainly focuses on viewpoint mismatches between domains and relies on time-synchronized demonstrations with simultaneous viewpoints. Chae et al. [7] introduced an IL algorithm designed to train a policy that exhibits robustness against variations in environment dynamics by imitating multiple experts. In contrast, our method encompasses general domain mismatches including the differences in viewpoint, degree of freedom, dynamics, and embodiment.

We focus on solving domain shift IL with image observation. Liu et al. [28] used time-aligned expert demonstrations in multiple source domains and learned a model to transform the demonstration from one domain into another. Our work is closely related to Third-Person Imitation Learning (TPIL) [37], which trains a model based on domain-independent feature extraction [15] with GAIL [18]. Similarly, Cetin & Celiktutan [6] proposed restricting mutual information between non-expert data and domain labels. Our method improves on the idea of feature extraction from non-time-aligned visual data by applying dual feature extraction and image reconstruction with consistency checks.

**IL from observations (IfO)**   IfO is an area of IL, where the expert state-action trajectory is not provided explicitly. In IfO, the learner should capture the behavioral information from observation. Demonstrations can be provided either as a set of state vectors without actions [8, 39] or simply as image observations [13, 21, 28, 43]. Torabi et al. [39] learned the inverse dynamics to infer the missing actions from observations in the demonstration. Edwards et al. [8] used latent actions to model the transition between two consecutive observations and relabel those latent actions to true actions. We assume that the learner knows its own state and action but expert demonstration is provided only in the form of visual observations as in other works ([6, 37].

## 3 System Model

**Setup**   An Markov Decision Process (MDP) is described by a tuple $(\mathcal{S}, \mathcal{A}, P, R, \gamma, \rho_0)$, where $\mathcal{S}$ is the state space, $\mathcal{A}$ is the action space, $P : \mathcal{S} \times \mathcal{A} \times \mathcal{S} \rightarrow \mathbb{R}^+$ is the state transition probability, $R : \mathcal{S} \times \mathcal{A} \rightarrow \mathbb{R}$ is the reward function, $\gamma \in (0, 1)$ is the discount factor, and $\rho_0$ is the initial state distribution [38]. Given an MDP, a stochastic policy $\pi : \mathcal{S} \times \mathcal{A} \rightarrow \mathbb{R}^+$ is a conditional probability over actions given state $s \in \mathcal{S}$. The goal of RL is to find an optimal policy $\pi^*$ that maximizes

$J(\pi) = \mathbb{E}_\pi[\sum_{t=0}^\infty \gamma^t R(s_t, a_t)]$, where $s_0 \sim \rho_0, a_t \sim \pi(\cdot|s_t), s_{t+1} \sim P(\cdot|s_t, a_t)$. In IL, expert demonstrations are provided instead of the true $R$. The goal of IL with a single domain is to imitate the demonstration generated by an expert policy $\pi_E$.

**Problem formulation** We now define IL with domain adaptation. We have two domains: *source domain* and *target domain*. The source domain is where we can obtain expert demonstrations, and the target domain is where we train the agent. Both domains are modeled as MDPs. The MDPs of the source domain (using subscript $S$) and the target domain (using subscript $T$) are given by $\mathcal{M}_S = (\mathcal{S}_S, \mathcal{A}_S, P_S, R_S, \gamma_S, \rho_{0,S})$ and $\mathcal{M}_T = (\mathcal{S}_T, \mathcal{A}_T, P_T, R_T, \gamma_T, \rho_{0,T})$, respectively. An expert is the one who performs a task optimally. The agent in the target domain learns the task from expert demonstrations in the source domain. We call the agent a *learner*. The *expert policy* $\pi_E : \mathcal{S}_S \times \mathcal{A}_S \to \mathbb{R}^+$ is in the source domain and the *learner policy* $\pi_\theta : \mathcal{S}_T \times \mathcal{A}_T \to \mathbb{R}^+$ is in the target domain. We assume that a set of image observations of $\pi_E$ are provided, which is denoted by $O_{SE}$, where '$SE$' refers to 'source expert'. The set of observations generated by $\pi_\theta$ is denoted by $O_{TL}$, where '$TL$' refers to 'target learner'. In addition to expert demonstrations from the source domain, we assume access to the non-optimal trajectories of image observations from both the source and target domains. The visual observations of rollouts from a non-expert policy $\pi_{SN} : \mathcal{S}_S \times \mathcal{A}_S \to \mathbb{R}^+$ in the source domain are denoted by $O_{SN}$, and the visual observations of rollouts from a non-expert policy $\pi_{TN} : \mathcal{S}_T \times \mathcal{A}_T \to \mathbb{R}^+$ in the target domain are denoted by $O_{TN}$. Here, '$SN$' refers to 'source non-expert', and '$TN$' refers to 'target non-expert'. $O_{SN}$ and $O_{TN}$ are additional data to assist our feature extraction and image reconstruction process. $\pi_{SN}$ and $\pi_{TN}$ can be constructed in several ways, but we choose these policies as ones taking uniformly random actions. Each observation set has size $n_{demo}$ and each observation $o_t$ consists of 4 images corresponding to 4 consecutive timesteps. The goal is to train $\pi_\theta$ to solve a given task in the target domain using $O_{SE}$, $O_{SN}$, $O_{TN}$, and $O_{TL}$.

## 4 Motivation

IL with domain adaptation presents a major challenge as the learner cannot directly mimic the expert demonstration, especially when it is given in a visual form. Also, we cannot leverage expert demonstrations for proxy tasks. To overcome domain shift, conventional methods [6, 37] extracts domain-invariant features from images. We call such feature *behavior feature*, capturing only expert or non-expert behavior information. One example is TPIL [37]. TPIL consists of a single behavior encoder $BE$, a domain discriminator $DD$, and a behavior discriminator $BD$, as shown in Fig. 1 (The label $XB_X$ in Fig. 1 means that the input is in the $X$ domain with behavior $B_X$. $X$ can be source $S$ or target $T$, and $B_X$ can be expert $E$ or non-expert $N$. We use this notation throughout the paper. See Appendix C for more details). The key idea is that $BE$ learns to fool $DD$ by removing domain information from the input while helping $BD$ by preserving behavior information from the input.

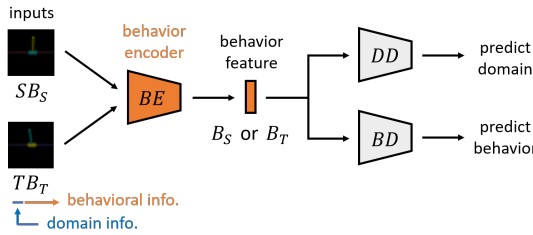

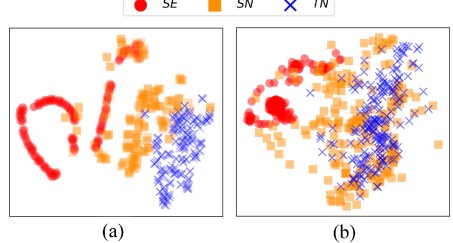

Figure 1: Basic structure for domain-independent behavior feature extraction: $BE$ - behavior encoder, $BD$ - behavior discriminator, $DD$ - domain discriminator

Figure 2: T-SNE plots of features from (a) TPIL and (b) DisentanGAIL. Each point represents a behavior feature: Red circle ($SE$), orange square ($SN$) and blue cross ($TN$)

However, the behavior feature extraction of the conventional methods does not work to our satisfaction. Fig. 2 shows the t-SNE [42] plots of the behavior features extracted by the conventional methods [6, 37] after training for the task of Reacher with two links (source domain) to three links (target domain). As seen in the figure, the behavior feature vectors of $SN$ (orange squares) and those of $TN$ (blue crosses) are separated in the left figure (note that when domain information is completely

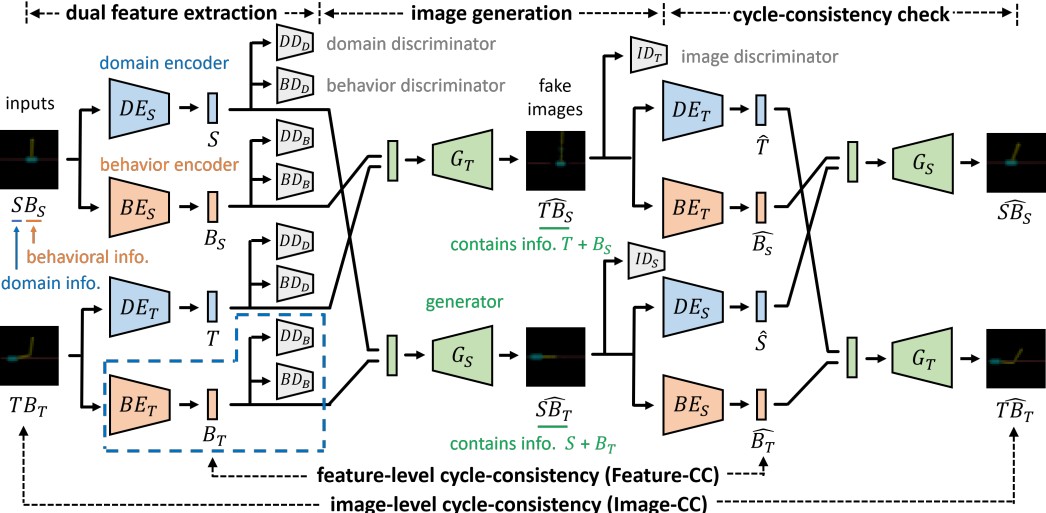

Figure 3: The structure of the learning model: Inputs in source and target domains have features $SB_S$ and $TB_T$, respectively. $B_S$ can be $E$ (expert) and $N$ (non-expert), and $B_T$ can be $N$ (non-expert) and $L$ (learner). $DE$ is domain encoder (blue), $BE$ is behavior encoder (orange) and $G$ is generator (green). $DD, BD$ and $ID$ with gray color are domain discriminator, behavior discriminator, and image discriminator, respectively. Small long rectangles represent features. The sub-block in the blue dotted line in the lower left corner corresponds to the basic structure shown in Fig. 1.

removed, the behavior feature vectors of $SN$ and $TN$ should overlap). Furthermore, since the target non-expert performs random actions for each given state, some actions resemble the expert actions in a given state, while other actions do not. So, there should be some overlap between the blue crosses ($TN$) and the red dots ($SE$). However, the blue crosses are well separated from the red circles in the right figure. Thus, the domain information is not properly removed from the behavior feature. Extracting domain-independent behavior features needs to fully exploit available source-domain and target-domain data.

Our approach is to embed the basic feature extraction structure into a bigger model with data translation. Our main idea for this is *dual feature extraction* and *dual cycle-consistency*. First, we propose a dual feature extraction scheme to extract both the domain feature and behavior feature from the input so that these two features are independent and contain full information about the input. Second, we propose a new cycle-consistency scheme that fits domain-adaptive IL with visual observation. The proposed model aims to extract better features for reward generation rather than to generate images. Thus, solely applying image-level cycle-consistency is not sufficient to solve the problem of our interest. We circumvent this limitation by additionally applying feature-level cycle-consistency on top of image-level cycle-consistency. Third, we adopt a reward-generating discriminator outside the feature extraction model for proper reward generation by exploiting the generated target-domain expert data from the feature extraction model. Therefore, we name our approach as *D3IL (Dual feature extraction and Dual cycle-consistency for Domain adaptive IL with visual observation)*.

## 5 Proposed Methodology

### 5.1 Dual Feature Extraction

The behavior feature in the target domain is the required information that tells whether the learner's action is good or not, in other words, whether the learner in the target domain successfully mimics the expert in the source domain or not. To improve behavior feature extraction, we boost the simple adversarial learning, as the structure shown in Fig. 1, by adding an additional structure from which we can check that the obtained feature well contains the required information without information loss, based on image reconstruction and consistency check, as shown in Fig. 3. That

is, we adopt two encoders in each of the source and target domains: domain encoder $DE_X$ and behavior encoder $BE_X$, where $X = S$ or $T$, as shown in the left side of Fig. 3. The domain encoder $DE_X$, where $X = S$ or $T$, tries to extract domain feature ($S$ or $T$) that contains only the domain information excluding behavioral information, whereas the behavior encoder $BE_X$ tries to extract behavioral feature ($B_X = E$ or $N$) that contains only the behavioral information (expert or non-expert) excluding domain information. Each of these encodings is based on the aforementioned basic adversarial learning block composed of one encoder and two accompanying discriminators: domain and behavior discriminators, as shown in the left side of Fig. 3. As the outputs of the four encoders, we have four output features: $S$ and $B_S$ from the source-domain encoders, and $T$ and $B_T$ from the target-domain encoders.

## 5.2 Dual Cycle-Consistency (Dual-CC)

We now explain *dual cycle-consistency (Dual-CC)*, composed of *image-level cycle-consistency (Image-CC)* and *feature-level cycle-consistency (Feature-CC)* to improve feature extraction. Using the four output feature vectors extracted by the encoders in Sec. 5.1, we apply image translation [19, 25, 26, 44, 48]. That is, we combine $T$ and $B_S$ and generate an image $\widehat{TB_S}$ by using generator $G_T$, and combine $S$ and $B_T$ and generate an image $\widehat{SB_T}$ by using generator $G_S$. Each image generator ($G_T$ or $G_S$) is trained based on adversarial learning with the help of the corresponding image discriminator ($ID_T$ or $ID_S$), as shown in the middle part of Fig. 3. Then, from the generated images $\widehat{TB_S}$ and image $\widehat{SB_T}$, we apply feature extraction again by using the same trained encoders $DE_T, BE_T, DE_S, BE_S$ used in the first-stage feature extraction, as shown in the right side of Fig. 3. Finally, using the four features $\hat{T}, \widehat{B}_S, \widehat{S}, \widehat{B}_T$ from the second-stage feature extraction, we do apply image translation again to reconstruct the original images $\widehat{SB_S}$ and $\widehat{TB_T}$ by using the same generators $G_S$ and $G_T$ used as the first-stage image generators, as shown in the rightmost side of Fig. 3. Note that if the encoders properly extracted the corresponding features without information loss, the reconstructed images $\widehat{SB_S}$ and $\widehat{TB_T}$ should be the same as the original input images $SB_S$ and $TB_T$. We exploit this *Image-CC* as one of our criteria for our feature-extracting encoders.

However, the aforementioned adversarial learning and Image-CC loss are insufficient to yield mutually exclusive behavior and domain features, as we will see in Sec. 6.7. To overcome this limitation, we enhance the dual feature extraction by complementing Image-CC by imposing *Feature-CC*. Note that our goal is not image translation but feature extraction relevant to domain-adaptive IL. The Feature-CC means that the extracted features $S, B_S, T$ and $B_T$ in the left dual feature extraction part of Fig. 3 should respectively be the same as their corresponding feature vectors $\hat{S}, \hat{B}_S, \hat{T}$ and $\hat{B}_T$ in the right cycle-consistency check part of Fig. 3. The rationale for this consistency is as follows. Consider $B_T$ and $\widehat{B_T}$ for example. $B_T$ is extracted from the target-domain image $TB_T$, whereas $\widehat{B_T}$ is extracted from a source-domain image $\widehat{SB_T}$ generated from source-domain image generator $G_S$. For the two behavior features $B_T$ and $\widehat{B_T}$ from two different domains to be the same, the domain information in $B_T$ and $\widehat{B_T}$ should be eliminated. Thus, the Feature-CC helps the extraction of mutually exclusive behavior and domain features while the Image-CC requires the combination of behavior and domain features to preserve full information about the input observation without any loss. Since both consistencies are not perfect, we adopt both Image-CC and Feature-CC for our feature extraction, so named Dual-CC.

## 5.3 Further Consistency and Stability Enhancement

In addition to the Dual-CC, our architecture allows us to verify other consistencies on the extracted features. First, we enforce the *image reconstruction consistency* by combining the features $S$ and $B_S$ from $DE_S$ and $BE_S$ in the first-stage feature extraction and feeding the feature combination $(S, B_S)$ into image generator $G_S$. The output $\widetilde{SB_S}$ should be the same as the original image $SB_S$. The same consistency applies to the feature combination $(T, B_T)$ in the target domain with image generator $G_T$. Second, we enforce the *feature reconstruction consistency* by feeding the generated source-domain image $\widetilde{SB_S}$ described above into the encoders $DE_S$ and $BE_S$, then we obtain domain feature $\tilde{S}$ and behavior feature $\tilde{B}_S$. These two features should be identical to the features $S$ and $B_S$ extracted in the first-stage feature extraction. The same principle applies to the target domain.

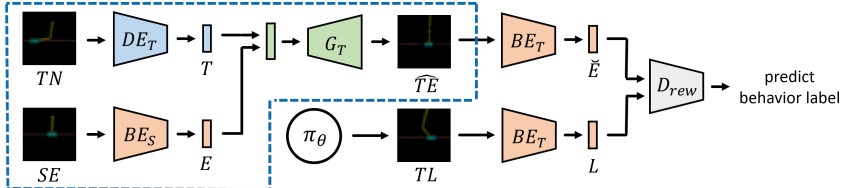

Figure 4: Reward-generating discriminator $D_{rew}$ and its training

In addition to the key consistency idea explained above, we apply further techniques of *feature vector regularization* and *feature vector similarity* to enhance performance. Feature vector regularization maintains the $L_2$-norm of the feature vector which helps to prevent the feature vectors from becoming too large. Feature vector similarity is another form of consistency that guides the model so that the domain feature vectors of observations from the same domain should have similar values and the behavior feature vectors of observations from the same behavior should have similar values.

With this additional guidance for feature extraction on top of the conventional adversarial learning block, the proposed scheme significantly improves feature extraction compared to previous approaches, as we will see in Sec. 6. Once the overall structure is learned, we take the behavior encoder $BE_T$ in the target domain (the block inside the blue dotted block in Fig. 3) to train the learner policy. This improved behavior feature extraction can enhance the learner's performance in the target domain. Note that the complexity of D3IL is simply at the level of widely used image translation.

## 5.4 Reward Generation and Learner Policy Update

Now, let us explain how to update the learner policy $\pi_\theta$. One can simply use the behavior discriminator $BD_B$ in the target domain together with the behavior encoder $BE_T$ to generate a reward for policy updates. Instead, we propose a different way of reward generation. We use $BE_T$ but do not use $BD_B$ from the feature extraction model. This is because, in our implementation, $BD_B$ is trained to distinguish the expert behavior $E$ from the non-expert behavior $N$, but what we need is a discriminator that distinguishes the expert behavior $E$ from the *learner* behavior $L$ so that the *learner* policy $\pi_\theta$ can receive proper rewards for imitation. Therefore, we train another network called *reward-generating discriminator $D_{rew}$* for reward generation. $D_{rew}$ and $\pi_\theta$ jointly learn in an adversarial manner like in GAIL. $D_{rew}$ takes a behavior feature vector from $BE_T$ as input and learns to predict the behavior label ($E$ or $L$) of the input. For stable learning, we consider GAIL with gradient penalty (GP) [24].

To train $D_{rew}$, we need the expert behavior feature from $BE_T$ as well as the learner behavior feature $L$. $L$ is simply obtained by applying the policy output $o_{TL}$ to $BE_T$. The required expert behavior feature can be obtained by applying generated target-domain expert image $\widehat{TE}$ to $BE_T$, where $\widehat{TE}$ is generated from the feature extraction model using domain feature $T$ from $o_{TN}$ and behavior feature $E$ from $o_{SE}$, as shown in Fig. 4. (Or simply we can use the expert feature $E$ obtained by applying source-domain expert image $o_{SE}$ to $BE_S$ based on the imposed Feature-CC.) The loss for $D_{rew}$ with fixed behavior encoder $BE_T$ is given by

$$L_D = \mathbb{E}_{(o_{TN}, o_{SE}) \in (O_{TN}, O_{SE})}[\log(D_{rew}(BE_T(\hat{o}_{TE})))] + \mathbb{E}_{o_{TL} \in O_{TL}}[\log(1 - D_{rew}(BE_T(o_{TL})))]$$
$$+ \mathbb{E}_{o_x \in Mix(\hat{o}_{TE}, o_{TL})}[(\|\nabla D_{rew}(BE_T(o_x))\|_2 - 1)^2]$$

where $\hat{o}_{TE} = G(DE(o_{TN}), BE(o_{SE}))$ is a generated image from the domain feature $T$ of $o_{TN}$ and the behavior feature $E$ of $o_{SE}$, and the third expectation is the GP term and is over the mixture of $\hat{o}_{TE}$ and $o_{TL}$ with a certain ratio. $D_{rew}$ takes a behavior feature vector as input and predicts its behavior label $E$ or $L$ by assigning value 1 to expert behavior and value 0 to learner behavior. On the other hand, $\pi_\theta$ learns to generate observations $o_{TL}$ so that the corresponding behavior feature vector resembles that of the expert. The learner updates its policy using SAC [17], and the estimated reward for an observation $o_t$ is defined by $\hat{r}(o_t) = \log(D_{rew}(BE_T(o_t))) - \log(1 - D_{rew}(BE_T(o_t)))$. Appendix A provides the details of loss functions implementing D3IL, and Algorithms 1 and 2 in Appendix B summarise our domain-adaptive IL algorithm.

# 6 Experiments

## 6.1 Experimental Setup

We evaluated D3IL on various types of domain shifts in IL tasks, including changes in visual effects, the robot's degree of freedom, dynamics, and embodiment. We also tested if D3IL can solve tasks when obtaining expert demonstrations by direct RL in the target domain is challenging. Each IL task consists of a source domain and a target domain, with a base RL task either in Gym [5] or in DeepMind Contol Suite (DMCS) [41]. In Gym, we used five environments: Inverted Pendulum (IP), Inverted Double Pendulum (IDP), Reacher-two (RE2), Reacher-three (RE3), and HalfCheetah (HC). In addition, we used a customized U-Maze environment [20] for the experiment in Section. 6.6. In DMCS, we used three environments: CartPole-Balance, CartPole-SwingUp, and Pendulum. Appendix E contains sample observations, task configuration, and space dimensions for each environment.

We compared the performance of D3IL with three major baselines: TPIL [37] and DisentanGAIL [6], and GWIL [9]. TPIL and DisentanGAIL are image-based IL methods that aim to extract domain-independent features from image observations and are most relevant to our problem setting. Additionally, we also compared D3IL with GWIL, a recent state-based IL method. For GWIL, demonstrations were provided as state-action sequences for all experiments, while for other methods demonstrations were provided as image observations. Appendix D provides implementation details including network structures and hyperparameters for D3IL.

## 6.2 Results on IL Tasks with Changing Visual Effects

We evaluated D3IL on tasks where the domain difference is caused by visual effects on image observations. We conducted the experiment in four IL tasks: IP and IDP, with different color combinations of the pole and cart (IP-to-colored and IDP-to-colored tasks), and RE2 and RE3, with different camera angles (RE2-to-tilted and RE3-to-tilted tasks). Figure 17 in Appendix F.1 contains sample images for each task. The average episodic return for D3IL and the baselines over 5 seeds are shown in Fig. 18 in Appendix F.1. As shown in Fig. 18, D3IL outperformed the baselines with large margins on tasks with changing visual effects.

## 6.3 Results on IL Tasks with Changing Degree-of-Freedom of a Robot

Next, we evaluated D3IL on IL tasks with varying robot's degree-of-freedom. This includes scenarios where the learner robot should imitate an expert performing the same task, but the number of joints differs between them. We conducted four IL tasks: two based on IP (one pole) and IDP (two poles), and two based on RE2 (2-link arm) and RE3 (3-link arm). Sample observations for each IL task are shown in Figure 14 in Appendix E. Fig. 5 shows the average episodic return over 5 seeds. It is seen that D3IL outperforms the baseline methods by a large margin in all the IL tasks. In Fig. 5 the dotted line represents the performance of SAC trained with true rewards in the target domain. This dotted line can be considered as the performance upper bound when using SAC as the control algorithm. It is interesting to observe that in Figs. 5 (c) and (d), the baselines seem to learn the task initially but yield gradually decreasing performance, whereas D3IL continues to improve its performance. Appendix F.2 provides further discussions on the quality of extracted features on these tasks. Appendix F.3 provides additional results on IP, IDP, RE2, and RE3 with simpler settings created by [6].

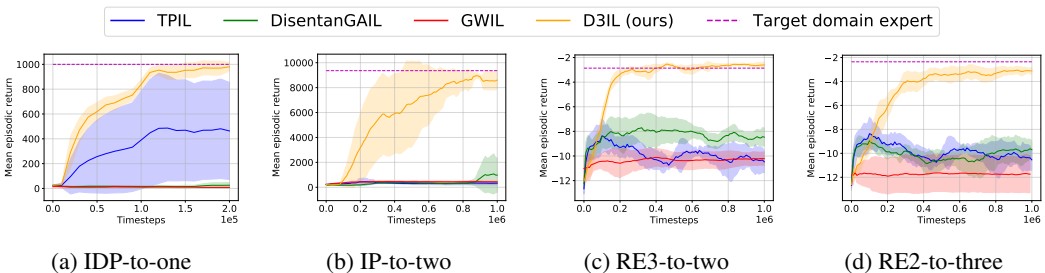

|          |          |          |          |
|----------|----------|----------|----------|
| (a) IDP-to-one | (b) IP-to-two | (c) RE3-to-two | (d) RE2-to-three |

Figure 5: The learning curves in the target domain of the IL tasks with changing DOF

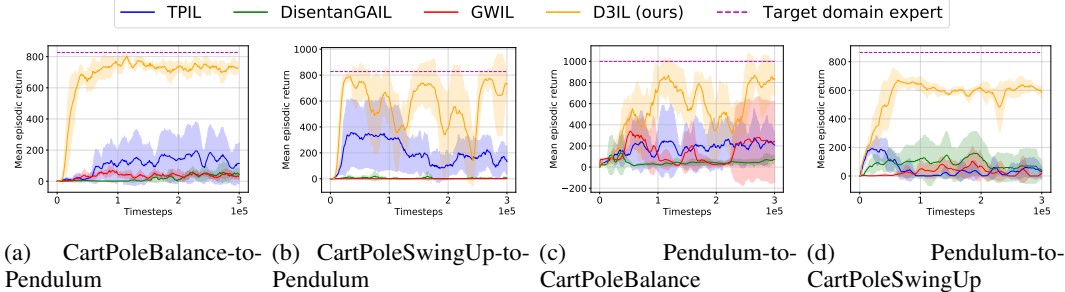

(a) CartPoleBalance-to-Pendulum  (b) CartPoleSwingUp-to-Pendulum  (c) Pendulum-to-CartPoleBalance  (d) Pendulum-to-CartPoleSwingUp

Figure 7: The learning curves for the considered IL tasks for CartPole and Pendulum environments

## 6.4 Results on Tasks with Changing Internal Dynamics

We examined D3IL on a task with a different type of domain shift, specifically a change in the robot's dynamics. The considered IL task is 'HalfCheetah-to-LockedFeet'. The base RL task is as follows. In HalfCheetah (HC) environment, the goal is to move an animal-like 2D robot forward as fast as possible by controlling six joints. Similar to [6], we also have a modified environment named as 'HalfCheetah-LockedFeet (HC-LF)'. In HC-LF, the robot can move only four joints instead of six, as two feet joints are immobilized. Fig. 15 in Appendix E shows sample observations for both environments, where the immobilized feet are colored in red. Fig. 6 shows the average episodic return over 5 seeds. Fig. 6 shows that D3IL has superior performance to the baseline methods and can solve IL tasks by changing robot's dynamics.

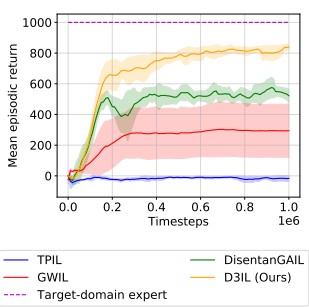

Figure 6: Result on HC-LF

## 6.5 Results on Tasks with Changing Robot Embodiment

We examined whether D3IL can train the agent to mimic an expert with a different embodiment. We use CartPole-Balance, CartPole-SwingUp, and Pendulum environments from DMCS [41], where sample observations are provided in Fig. 16 (see Appendix E). The goal for all environments is to keep the pole upright. The CartPole agents do so by sliding the cart, while the Pendulum agent does so by adding torque on the center. The pole is initially set upright in CartPole-Balance, while it is not in other environments. Domain adaptation between these tasks is challenging because not only the embodiment between agents in both domains are different but also the distributions of expert and non-expert demonstrations, and initial observations for both domains are quite different. Fig. 7 shows the performances on IL tasks with different robot embodiment. Although D3IL does not fully reach the upper bound, it achieved far better performance than the previous methods on tasks by changing the robot's embodiment.

## 6.6 Experiment on AntUMaze: When Direct RL in Target Domain is Difficult

In fact, for IL tasks we have done so far, the learner policy can be learned directly in the target domain by RL interacting with the target-domain environment. The real power of domain-adaptive IL comes when such direct RL in the target domain is very difficult or impossible by known RL algorithms. Here, we consider one such case known as AntUMaze [20]. In AntUMaze, a quadruped ant robot should move to the red goal position through a U-shaped maze, as shown in the second row of Fig. 8. It can be verified that it is very difficult to solve this problem directly with a current state-of-the-art RL algorithm such as SAC. This is because the problem involves complicated robot manipulation and pathfinding simultaneously. In order to solve AntUMaze, we consider a simpler task in the source domain known as PointUMaze, which is easily solvable by SAC. In PointUMaze, the robot is simplified as a point whose state is defined as the point location and point direction, as shown in the first row of Fig. 8, whereas the state of the ant robot is much more complicated. Appendix D.3 provides a detailed explanation of the training process for the UMaze environments. Now, the goal

of this IL task (named PointUMaze-to-Ant) is to solve the AntUMaze problem in the target domain with its own dynamics based on the point robot demonstrations.

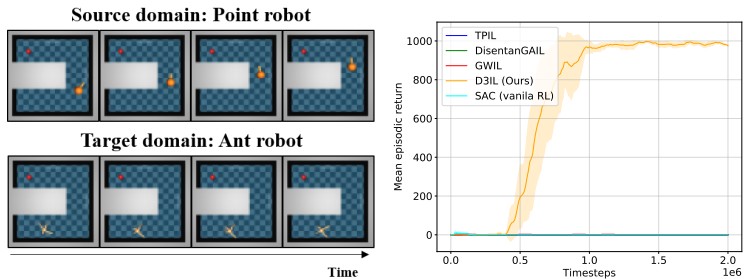

Figure 8: Sample images of U-maze

Figure 9: Results on PointUMaze-to-Ant

Fig. 9 shows the average episodic return over 5 seeds. As shown in Fig. 9, the ant robot trained with D3IL reaches the goal position after about 1M timesteps, while the ant robot trained with the baseline IL algorithms failed to reach the goal position. D3IL can train the ant agent to learn desirable behaviors from point-robot expert demonstrations with totally different dynamics. We observed that SAC with reward could not solve this task. So, it is difficult to directly solve the AntUMaze with RL or to obtain expert demonstrations in AntUMaze directly. D3IL can transfer expert knowledge from an easier task to a harder task, and train the agent in the target domain to perform the AntUMaze task.

## 6.7    Analysis of Dual Cycle-Consistency

To analyze the impact of Dual-CC, we use t-SNE to visualize behavior features generated by our model. Fig. 10 shows the t-SNE plots of the behavior feature extracted by the model trained on the RE2-to-three task: We used the model with full Dual-CC in Fig. 10 (a), the model with only Feature-CC in Fig. 10 (b), and the model with only Image-CC in Fig. 10 (c). Fig. 10 (a) based on the Dual-CC shows the desired properties; target-domain non-expert ($TN$) and source-domain non-expert ($SN$) are well overlapped, and some of the random behavior feature vectors coincide with the source-domain expert ($SE$) behavior. We also plotted behavior features of generated target experts ($\widehat{TE}$) with green '+' in the figure. Since the behavior feature of $\widehat{TE}$ is derived from the behavior feature of $SE$ and we imposed the Feature-CC, the red circles and green '+' overlap quite a good portion. However, there still exists some gap at a certain region, showing the feature extraction is not perfect. Fig. 10 (c) shows that domain information still remains when removing the Feature-CC, and the Feature-CC indeed helps eliminate domain information. As seen in Fig. 10 (b), only the Feature-CC is not enough to extract the desired behavior feature. Indeed, Dual-CC is essential in extracting domain-independent behavior features.

## 6.8    Ablation Study

**Impact of each loss component**  D3IL uses multiple add-ons to the basic adversarial behavior feature learning for consistency check. As described in Sec. 5, the add-ons for better feature extraction are (1) image and feature reconstruction consistency with the generators in the middle part of Fig. 3, (2) Image-CC with the right part of Fig. 3, (3) feature similarity, and (4) Feature-CC. We removed each component from the full algorithm in the order from (4) to (1) and

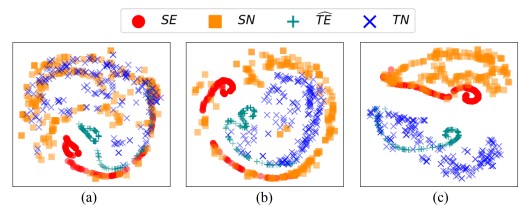

Figure 10: T-SNE plot of behavior feature from D3IL: (a) Dual-CC (b) Feature-CC only (c) Image-CC only

evaluated the corresponding performance, which is shown in Table 1. In Table 1, each component has a benefit to performance enhancement. Especially, the consistency check for the extracted feature significantly enhances the performance. Indeed, our consistency criterion based on dual feature extraction and image reconstruction is effective.

**Reward-generating discriminator** $D_{rew}$  In Sec. 5.4, we explained the use of $D_{rew}$ for reward estimation. As explained, we train $D_{rew}$ and the learner policy $\pi_\theta$ in an adversarial manner. Instead of using an external $D_{rew}$, we could use the behavior discriminator $BD_B$ in the feature extraction model to train the policy $\pi_\theta$. To see how this method works, we trained $\pi_\theta$ using the behavior discriminator $BD_B$ in the feature extraction model, not the external $D_{rew}$. As shown in Table 2, training $\pi_\theta$ using $BD_B$ is not successful because of the reason mentioned in Sec. 5.4.

Table 1: Impact of each loss component

|  | RE3-to-two |
|---|---|
| basic adversarial learning | $-10.7 \pm 0.5$ |
| + image/feature reconstr. consistency | $-6.0 \pm 0.7$ |
| + image-level cycle-consistency | $-3.0 \pm 0.1$ |
| + feature similarity | $-2.8 \pm 0.1$ |
| + feature-level cycle-consistency | **$-2.6 \pm 0.1$** |

Table 2: Reward-generating discriminator: $D_{rew}$ versus $BD_B$

|  | IP-to-color | RE2-to-tilted |
|---|---|---|
| Use $D_{rew}$ | **$1000 \pm 0$** | **$-4.3 \pm 0.3$** |
| Use $BD_B$ | $293 \pm 324$ | $-11.9 \pm 0.8$ |

**Domain encoders**  As described in Sec. 5, we proposed employing both domain encoder and behavior encoder for enhanced domain-independent behavior feature extraction. This approach was motivated by the observation that solely relying on a behavior encoder was insufficient to eliminate domain-related information from behavior features. To investigate the importance of domain encoders, we evaluated the performance of D3IL with and without domain encoders ($DE_S$ and $DE_T$) and their corresponding discriminators ($DD_D$ and $BD_D$) across four tasks: IP-to-two, RE2-to-three, RE3-to-tilted, and CartPoleBalance-to-Pendulum. All other networks and losses remain unchanged. Fig. 11 shows the results. The results demonstrate the benefit of conditioning generators on learned domain encoders to boost model performance.

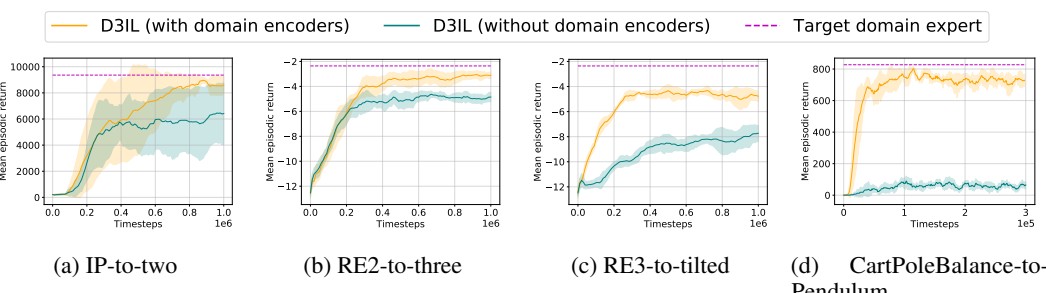

(a) IP-to-two    (b) RE2-to-three    (c) RE3-to-tilted    (d) CartPoleBalance-to-Pendulum

Figure 11: Ablation study on domain encoders

## 7 Discussion and Conclusion

In this paper, we have considered the problem of domain-adaptive IL with visual observation. We have proposed a new architecture based on dual feature extraction and image reconstruction in order to improve behavior feature extraction, and a reward generation procedure suited to the proposed model. The proposed architecture imposes additional consistency criteria on feature extraction in addition to basic adversarial behavior feature extraction to obtain better behavior features. Numerical results show that D3IL yields superior performance to existing methods in IL tasks with various domain differences and is useful when direct RL in the target domain is difficult.

**Limitations**  While our method demonstrates superior performance, it also has some limitations. One is the tricky tuning of loss coefficients, which involves several loss functions. However, we observed that balancing the scale of loss components is sufficient. Additionally, our feature extraction model remains fixed during policy updates. Future works could leverage the past target learner experiences to update the model further. Another limitation is the need for target environment interaction during training, a characteristic shared by several domain-adaptive IL approaches. It will be interesting to extend our method to an offline approach in future works. Moreover, our method lacks a quantitative measure for domain shifts. Future research could develop a mathematical way to quantify domain shifts, enabling the measurement of task difficulty and addressing more complex IL problems. Lastly, future works could extend our method to multi-task or multi-modal scenarios.

## Acknowledgments and Disclosure of Funding

This work was supported in part by Institute of Information & Communications Technology Planning & Evaluation (IITP) grant funded by the Korea government (MSIT) (No.2022-0-00469, Development of Core Technologies for Task-oriented Reinforcement Learning for Commercialization of Autonomous Drones) and in part by Institute of Information & Communications Technology Planning & Evaluation (IITP) grant funded by the Korea government (MSIT) (No.2022-0-00124, Development of Artificial Intelligence Technology for Self-Improving Competency-Aware Learning Capabilities). Prof. Seungyul Han is currently with Artificial Intelligence Graduate School of UNIST, and his work is partly supported by Artificial Intelligence Graduate School support (UNIST), IITP grant funded by the Korea government (MSIT) (No.2020-0-01336).

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

# A    The Loss Functions for the Proposed Model

In Sec. 5, we provided the key idea of the proposed learning model. Here, we provide a detailed explanation of the loss functions implementing the learning model described in Fig. 3 of the main paper. Table 3 shows the notations for observation sets.

Table 3: The notations for observation sets

|  | Notation |
|---|---|
| Source domain | $O_S = O_{SE} \cup O_{SN}$ |
| Target domain | $O_T = O_{TN} \cup O_{TL}$ |
| Expert behavior | $O_E = O_{SE}$ |
| Non-expert behavior | $O_N = O_{SN} \cup O_{TN}$ |
| All observations | $O_{ALL} = O_{SE} \cup O_{SN} \cup O_{TN} \cup O_{TL}$ |

The observation sets $O_{SE}, O_{SN}, O_{TN}$ and $O_{TL}$ are defined in Sec. 3 of the main paper.

## A.1    Dual Feature Extraction

The left side of Fig. 3 shows the basic adversarial learning block for dual feature extraction. That is, we adopt two encoders in each of the source and target domains: *domain encoder* and *behavior encoder*. Behavior encoders ($BE_S$, $BE_T$) learn to obtain *behavior feature vectors* from images that only preserve the behavioral information and discard the domain information. Here, subscript $S$ or $T$ denotes the domain from which each encoder takes input. *Domain encoders* ($DE_S$, $DE_T$) learn to obtain *domain feature vectors* which preserve only the domain information and discard the behavioral information. The encoders learn to optimize the WGAN objective [2, 16] based on the *feature discriminators*: behavior discriminator $BD_*$ and domain discriminator $DD_*$. Discriminators $BD_B$ and $DD_B$ are associated with the behavior encoder $BE_X$, where $X = S$ or $T$, and Discriminators $BD_D$ and $DD_D$ are associated with the domain encoder $DE_X$, where $X = S$ or $T$. $BD_B$ learns to identify the *behavioral* information from behavior feature vectors, and $DD_B$ learns to identify the *domain* information from behavior feature vectors. $BD_D$ learns to identify the *behavioral* information from domain feature vectors, and $DD_D$ learns to identify the *domain* information from domain feature vectors. For notational simplicity, we define functions $BE$ and $DE$ as

$$BE(o) = \begin{cases} BE_S(o) & \text{if } o \in O_S \\ BE_T(o) & \text{if } o \in O_T \end{cases} \tag{1}$$

$$DE(o) = \begin{cases} DE_S(o) & \text{if } o \in O_S \\ DE_T(o) & \text{if } o \in O_T. \end{cases} \tag{2}$$

Two loss functions are defined for the dual feature extraction: *feature prediction loss* and *feature adversarial loss*.

1) *Feature prediction loss:* This loss makes the model retain the desired information from feature vectors and is given by

$$L_{feat}^{pred} = -\Bigg( \mathbb{E}_{o \in O_E}[BD_B(BE(o))] - \mathbb{E}_{o \in O_N}[BD_B(BE(o))]$$

$$+ \mathbb{E}_{o \in O_S}[DD_D(DE(o))] - \mathbb{E}_{O \in O_T}[DD_D(DE(o))] \Bigg) \tag{3}$$

where $\mathbb{E}_{o \in A}[\cdot]$ is the expectation over inputs $o \in A$. The first term $\mathbb{E}_{o \in O_E}[BD_B(BE(o))]$ in the right-hand side (RHS) of (3) means that if the input observation of the encoder $BE(\cdot)$ is from the expert observation set $O_E$, then the discriminator $BD_B$ with the extracted feature should declare a high value. The second term $-\mathbb{E}_{o \in O_N}[BD_B(BE(o))]$ in the right-hand side (RHS) of (3) means that if the input observation of the encoder $BE(\cdot)$ is from nonexpert observation set $O_N$, then the discriminator $BD_B$ with the extracted feature should declare a low value. The third term $\mathbb{E}_{o \in O_S}[DD_D(DE(o))]$ in the right-hand side (RHS) of (3) means that if the input observation of the encoder $DE(\cdot)$ is from the source-domain set $O_S$, then the discriminator $DD_D$ with the extracted feature should declare a high

value. The fourth term $-\mathbb{E}_{o\in O_T}[DD_D(DE(o))]$ in the right-hand side (RHS) of (3) means that if the input observation of the encoder $DE(\cdot)$ is from the target-domain set $O_T$, then the discriminator $DD_D$ with the extracted feature should declare a low value. Since we define a loss function, we have the negative sign in front of the RHS of (3). (Note that $BD_B$ learns to assign higher values to expert behavior and to assign lower values to non-expert behavior. $DD_D$ learns to assign higher values to the source domain and to assign lower values to the target domain.) The encoders ($BE$, $DE$) help discriminators ($BD_B$, $DD_D$) to identify the behavioral or domain information from the input, and they jointly learn to minimize $L_{feat}^{pred}$.

2) *Feature adversarial loss:* This loss makes the model delete the undesired information from feature vectors by adversarial learning between the encoders and the discriminators. The loss is given by

$$
\begin{aligned}
L_{feat}^{adv} =&\mathbb{E}_{o\in O_S}[DD_B(BE(o))] - \mathbb{E}_{o\in O_T}[DD_B(BE(o))] \\
&+ \mathbb{E}_{o\in O_E}[BD_D(DE(o))] - \mathbb{E}_{o\in O_N}[BD_D(DE(o))].
\end{aligned}
\tag{4}
$$

In this loss function, the output of the behavior encoder is fed to the domain discriminator, and the output of the domain encoder is fed to the behavior discriminator. The learning of the encoders and the discriminators is done in an adversarial manner as

$$
\min_{BE,DE} \ \max_{DD_*,BD_*} \ L_{feat}^{adv}.
\tag{5}
$$

Recall that $BD_*$ learns to assign higher values to expert behavior and to assign lower values to non-expert behavior, and $DD_*$ learns to assign higher values to the source domain and to assign lower values to the target domain. Thus, the discriminators try to do their best, whereas the encoders try to fool the discriminators.

The first two terms in the RHS of (4) imply that the output of the behavior encoder $BE$ should *not* contain the domain information. The last two terms in the RHS of (4) imply that the output of the domain encoder $DE$ should *not* contain the behavioral information.

## A.2   Image Reconstruction and Associated Consistency Check

In the previous subsection, we explained the loss functions of basic adversarial learning for dual feature extraction. Now, we explain the loss functions associated with consistency checks with image reconstruction.

As shown in the middle part of Fig. 3, for image reconstruction, we adopt a *generator* in each of the source and target domains: generator $G_S$ for the source domain and generator $G_T$ for the target domain. *Generators* $G_S$ and $G_T$ learn to produce images from the feature vectors so that the generated images should resemble the original input images. $G_S$ takes a source-domain feature vector and a behavior feature vector as input and generates an image that resembles images in the source domain. On the other hand, $G_T$ takes a target-domain feature vector and a behavior feature vector as input and generates an image that resembles images in the target domain.

Generators $G_S$ and $G_T$ learn to optimize the WGAN objective with the help of image discriminators: $ID_S$ and $ID_T$. $ID_S$ distinguishes real and generated images in the source domain, and $ID_T$ does so in the target domain. For notational simplicity, we define functions $G(\cdot,\cdot)$ and $ID(\cdot,\cdot)$ as

$$
G(DE(o), BE(o')) = \begin{cases} G_S(DE(o), BE(o')) & \text{if } o \in O_S \\ G_T(DE(o), BE(o')) & \text{if } o \in O_T \end{cases}
\tag{6}
$$

$$
ID(o) = \begin{cases} ID_S(o) & \text{if } o \in O_S \\ ID_T(o) & \text{if } o \in O_T. \end{cases}
\tag{7}
$$

1) *Image adversarial loss*: The basic loss function for training the image generators and the associated blocks is image adversarial loss of WGAN learning [2, 16]. The image adversarial loss makes the model generate images that resemble the real ones and is given by

$$
L_{img}^{adv} = l^a(SE) + l^a(SN) + l^a(TN) + l^a(TL).
\tag{8}
$$

where $l^a(x)$ is the adversarial WGAN loss for a real image in $O_x$. In more detail, the first term in the RHS of (8) can be expressed as

$$l^a(SE) = \mathbb{E}_{o \in O_{SE}}[ID(o)] - \mathbb{E}_{(o_x, o_y) \in (O_{SN}, O_{SE})}[ID(G(DE(o_x), BE(o_y)))]. \qquad (9)$$

Note that the image discriminator $ID_*$ learns to assign a high value for a true image and a low value for a fake image. The first term in the RHS of (9) trains the image discriminator $ID(\cdot)$ to assign a high value to a true image $o \in O_{SE}$, whereas the second term in the RHS of (9) trains the image discriminator $ID(\cdot)$ to assign a low value to a fake image. The latter is because the $ID(\cdot)$ input, i.e., the generator output $G(DE(o_x), BE(o_y))$ with $(o_x, o_y) \in (O_{SN}, O_{SE})$ inside the second term in the RHS of (9) is a fake image; for this image, the behavior feature is taken from $o_y \in O_{SE}$ but the domain feature is taken from $o_x \in O_{SN}$ not in $O_{SE}$.

The other terms $l^a(SN)$, $l^a(TN)$ and $l^a(TL)$ in the RHS of (8) are similarly defined as

$$l^a(SN) = \mathbb{E}_{o \in O_{SN}}[ID(o)] - \mathbb{E}_{(o_x, o_y) \in \hat{O}_{SN, fake}}[ID(G(DE(o_x), BE(o_y)))] \qquad (10)$$

$$l^a(TN) = \mathbb{E}_{o \in O_{TN}}[ID(o)] - \mathbb{E}_{(o_x, o_y) \in \hat{O}_{TN, fake}}[ID(G(DE(o_x), BE(o_y)))] \qquad (11)$$

$$l^a(TL) = \mathbb{E}_{o \in O_{TL}}[ID(o)] - \mathbb{E}_{(o_x, o_y) \in \hat{O}_{TL, fake}}[ID(G(DE(o_x), BE(o_y)))] \qquad (12)$$

where $\hat{O}_{SN, fake}$ is the set of image combinations $\{(o_x, o_y)\}$ such that the generated fake image $G(DE(o_x), BE(o_y))$ has label $SN$. $\hat{O}_{TN, fake}$ and $\hat{O}_{TL, fake}$ are similarly defined. Specifically,

$$\hat{O}_{SN, fake} = (O_{SE}, O_{SN}) \cup (O_{SE}, O_{TN}) \cup (O_{SN}, O_{TN})$$

$$\hat{O}_{SN, fake} = (O_{SE}, O_{SN}) \cup (O_{SE}, O_{TN}) \cup (O_{SN}, O_{TN})$$

$$\hat{O}_{TL, pair} = (O_{TN}, O_{TL}).$$

With the image adversarial loss $L_{imag}^{adv}$ in (8), the learning is performed in an adversarial manner. The encoders and generators learn to minimize $L_{img}^{adv}$, while the image discriminators learn to maximize $L_{img}^{adv}$. That is, the image discriminators are trained to distinguish fake images from real images, whereas the encoders and the generators are trained to fool the image discriminators $ID_*$.

Note that the image adversarial loss is basically for training the image generators and the associated blocks in a WGAN adversarial manner. With the availability of the image generators $G_S$ and $G_T$, we can impose our first and second consistency criteria: *image reconstruction consistency* and *feature reconstruction consistency*. We define the corresponding loss functions below.

2) *Image reconstruction consistency loss*: This loss checks the feature extraction is properly done. When we combine the features $S$ and $B_S$ from $DE_S$ and $BE_S$ with an input true image $o_{SE}$ (or input true image $o_{SN}$) in the first-stage feature extraction and input the feature combination $(S, B_S)$ into image generator $G_S$, the generated image should be the same as the original observation image $o_{SE}$ (or $o_{SN}$). The same consistency applies to the feature combination $(T, B))$ in the target domain with a true input image $o_{TN}$ or $o_{TL}$ and the image generator $G_T$. Thus, the image reconstruction loss is given as

$$L_{img}^{recon} = \mathbb{E}_{o \in O_{ALL}}[l_{mse}(o, \hat{o})] \qquad (13)$$

where

$$\hat{o} = G(DE(o), BE(o)), \qquad (14)$$

$O_{ALL} = O_{SE} \cup O_{SN} \cup O_{TN} \cup O_{TL}$, and $l_{mse}(u, v)$ is the mean square error between $u$ and $v$. The encoders and generators learn to minimize $L_{img}^{recon}$.

3) *Feature reconstruction consistency loss*: This is the second self-consistency criterion. If we input the generated source-domain image $\widetilde{SB_S}$ described in the above image reconstruction loss part into the encoders $DE_S$ and $BE_S$, then we obtain domain feature $\tilde{S}$ and behavior feature $\tilde{B}_S$, and these two features should be the same as the features $S$ and $B_S$ extracted in the first-stage feature extraction.

The same principle applies to the target domain. Thus, the *feature reconstruction consistency loss* is expressed as

$$L_{feat}^{recon} = \mathbb{E}_{o \in O_{ALL}}[||BE(o) - BE(\hat{o})||_2^2] + \mathbb{E}_{o \in O_{ALL}}[||DE(o) - DE(\hat{o})||_2^2] \qquad (15)$$

where

$$\hat{o} = G(DE(o), BE(o)), \qquad (16)$$

$|| \cdot ||_2$ denotes the 2-norm of a vector, and $O_{ALL} = O_{SE} \cup O_{SN} \cup O_{TN} \cup O_{TL}$. Note that $\hat{o}$ is a generated image from the domain and behavior features extracted from $o$, and $DE(\hat{o})$ and $BE(\hat{o})$ are the domain and behavior features from the generated image $\hat{o}$. The encoders and generators learn to minimize $L_{feat}^{recon}$.

### A.3 Cycle-Consistency Check

Now, let us consider our third consistency criterion: *cycle-consistency check* with the right side of Fig. 3. This consistency check involves image translation in the middle part of Fig. 3 and the image retranslation in the right part of Fig. 3 in the main paper, and requires that the original image and the reconstructed image of translation/retranslation should be the same for perfect feature extraction and generation. The *image cycle-consistency loss* is expressed as

$$L_{img}^{cycle} = \mathbb{E}_{(o_x, o_y) \in O_{cycle}^2}[l_{mse}(o_x, G(DE(\hat{o}_{(x,y)}), BE(\hat{o}_{(y,x)})))] \qquad (17)$$

where

$$\hat{o}_{(x,y)} = G(DE(o_x), BE(o_y)) \qquad (18)$$

$$\hat{o}_{(y,x)} = G(DE(o_y), BE(o_x)) \qquad (19)$$

and $O_{cycle}^2$ is the set of image pairs such that the two images in a pair do not belong to the same domain, i.e., one image belongs to the source domain and the other image belongs to the target domain. The explanation is as follows. Consider that we apply image $o_x$ to the upper input and image $o_y$ to the lower input of the left side of Fig. 3 in the main paper. Then, $\hat{o}_{(x,y)} = G(DE(o_x), BE(o_y))$ is the generated image in the lower row and $\hat{o}_{(y,x)} = G(DE(o_y), BE(o_x))$ is the generated image in the upper row in the middle part of Fig. 3 in the main paper. Finally, the image $G(DE(\hat{o}_{(x,y)}), BE(\hat{o}_{(y,x)}))$ is the final reconstructed image in the upper row of the right side of Fig. 3 in the main paper. So, we can check the cycle-consistency between $o_x$ and $G(DE(\hat{o}_{(x,y)}), BE(\hat{o}_{(y,x)}))$. The situation is mirrored when we apply image $o_x$ to the lower input and image $o_y$ to the upper input of the left side of Fig. 3 in the main paper. The encoders and generators learn to minimize $L_{img}^{cycle}$.

The *feature cycle-consistency loss* is expressed as

$$L_{feat}^{cycle} = \mathbb{E}_{(o_x, o_y) \in O_{cycle}^2}[l_{mse}(DE(o_x), DE(\hat{o}_{(x,y)}))] + \mathbb{E}_{(o_x, o_y) \in O_{cycle}^2}[l_{mse}(BE(o_y), BE(\hat{o}_{(y,x)}))] \qquad (20)$$

where

$$\hat{o}_{(x,y)} = G(DE(o_x), BE(o_y)) \qquad (21)$$

The explanation is as follows. For example, from the target domain real image $TB_T$, we extract domain feature $T$ and behavior feature $B_T$. From these features, we can reconstruct $TB_T$. Also, we extract domain feature $\widehat{T}$ (from $\widehat{TB_S}$) and behavior feature $\widehat{B_T}$ (from $\widehat{SB_T}$). From these features, we can reconstruct $TB_T$. Let's assume $\widehat{B_T}$ (behavior feature from the source domain image $\widehat{SB_T}$ generated by $G_S$) is replaced by $B_T$ (behavior feature from the target domain image $TB_T$). Then from $\widehat{T}$ and $B_T$ we should still reconstruct $TB_T$. That is, $\widehat{B_T}$ (behavior feature from the source domain image $\widehat{SB_T}$ made by $G_S$) = $B_T$ (behavior feature from the target domain image $TB_T$). They are behavior features from different domains but they are equal. This implies that the behavior feature is independent of domain information. We name this by feature cycle-consistency because this constraint implicitly satisfies image cycle-consistency. if we explicitly constrain $\widehat{B_T} = B_T$ and $\widehat{T} = T$, then we can replace $\widehat{B_T}$ and $\widehat{T}$ by $B_T$ and $T$, and the image cycle-consistency is implicitly satisfied.

### A.4  Other Losses

We further define losses to enhance performance.

1) *Feature regularization loss*: The *feature regularization loss* prevents the feature vector values from exploding for stable learning, and is given by

$$L_{feat}^{reg} = \mathbb{E}_{o \in O_{ALL}}[||BE(o)||_2^2 + ||DE(o)||_2^2]. \tag{22}$$

Instead, more rigorously, we can use the following regularization:

$$
\begin{aligned}
L_{feat}^{reg,1} &= \mathbb{E}_{o \in O_S}[(||DE_S(o)|| - c_{norm,d})^2] \\
L_{feat}^{reg,2} &= \mathbb{E}_{o \in O_S}[(||BE_S(o)|| - c_{norm,b})^2] \\
L_{feat}^{reg,3} &= \mathbb{E}_{o \in O_T}[(||DE_T(o)|| - c_{norm,d})^2] \\
L_{feat}^{reg,4} &= \mathbb{E}_{o \in O_T}[(||BE_T(o)|| - c_{norm,b})^2] \\
L_{feat}^{reg} &= L_{feat}^{reg,1} + L_{feat}^{reg,2} + L_{feat}^{reg,3} + L_{feat}^{reg,4}.
\end{aligned} \tag{23}
$$

Where, $c_{norm,d}$ and $c_{norm,b}$ are hyperparameters with non-negative values. Appendix D.2 provides details including the value of $c_{norm,d}$ and $c_{norm,b}$ for the experiment.

2) *Feature similarity loss*:  This loss aims to map observations of the same domain to similar points in the feature vector space and to map observations of the same behavior to similar points in the feature vector space. The feature similarity loss is defined as

$$L_{feat}^{sim} = ||\mu_{BE(O_{SN})} - \mu_{BE(O_{TN})}||_2^2 + ||\mu_{DE(O_{SE})} - \mu_{DE(O_{SN})}||_2^2 + ||\mu_{DE(O_{TN})} - \mu_{DE(O_{TL})}||_2^2, \tag{24}$$

where $\mu_{BE(O_x)}$ and $\mu_{DE(O_x)}$ are the means of the behavior feature vectors and the domain feature vectors over the observation set $O_x$ ($x \in \{SE, SN, TN, TL\}$), respectively.

### A.5  Final Loss

Combining the above individual loss functions, we have the final objective for the encoders $E = (BE_S, DE_S, BE_T, DE_T)$, the generators $G = (G_S, G_T)$, feature discriminators $FD = (BD_B, DD_B, BD_D, DD_D)$ and image discriminators $ID = (ID_S, ID_T)$ as follows:

$$
\begin{aligned}
\min_{E,G} L_{E,G} = {} & \lambda_{feat}^{adv} L_{feat}^{pred} + \lambda_{feat}^{adv} L_{feat}^{adv} + \lambda_{img}^{adv} L_{img}^{adv} + \lambda_{img}^{recon} L_{img}^{recon} + \lambda_{feat}^{recon} L_{feat}^{recon} \\
& + \lambda_{img}^{cycle} L_{img}^{cycle} + \lambda_{feat}^{cycle} L_{feat}^{cycle} + \lambda_{feat}^{sim} L_{feat}^{sim} + \lambda_{feat}^{reg} L_{feat}^{reg}
\end{aligned} \tag{25}
$$

$$\min_{FD,ID} L_{FD,ID} = \lambda_{feat}^{adv} L_{feat}^{pred} - \lambda_{feat}^{adv} L_{feat}^{adv} - \lambda_{img}^{adv} L_{img}^{adv} \tag{26}$$

where the weighting factors $\lambda_{feat}^{adv}, \lambda_{feat}^{adv}, \lambda_{img}^{adv}, \lambda_{img}^{recon}, \lambda_{feat}^{recon}, \lambda_{img}^{cycle}, \lambda_{feat}^{cycle}, \lambda_{feat}^{sim}, \lambda_{feat}^{reg}$ are hyperparameters and their values are shown in Appendix D. Note that $L_{feat}^{adv}$ and $L_{img}^{adv}$ appear in both (25) and (26) with negative sign in (26). Hence, these two terms induce adversarial learning in the overall learning process.

### A.6  Reward Generation and Learner Policy Update

In this section, we explain the simple method of extracting expert features and training $D_{rew}$ for reward generation. That is, we just use the output of $BE_S$ in the source domain with expert input $o_{SE}$. In this case, $D_{rew}$ learns to minimize the following objective while behavior encoder $BE_T$ is fixed:

$$
\begin{aligned}
L_D = {} & \mathbb{E}_{o_{SE} \in O_{SE}}[\log(D_{rew}(BE(o_{SE})))] + \mathbb{E}_{o_{TL} \in O_{TL}}[\log(1 - D_{rew}(BE(o_{TL})))] \\
& + \mathbb{E}_{o_x \in Mix(\hat{o}_{TE}, o_{TL})}[(||\nabla D_{rew}(BE(o_x))||_2 - 1)^2].
\end{aligned} \tag{27}
$$

where the third expectation in (27) is over the mixture of $\hat{o}_{TE}$ and $o_{TL}$ with a certain ratio. The proposed method other than this simple method is explained in Sec. 5.
$D_{rew}$ takes a behavior feature vector as input and predicts its behavior label $E$ or $L$ from the input

behavior feature vector. $D_{rew}$ learns to assign the value 1 to the expert behavior and 0 to the learner behavior. On the other hand, $\pi_\theta$ learns to generate observations $O_{TL}$ so that the corresponding behavior feature vector looks like the expert. The learner updates the policy using SAC [17], and the estimated reward for an observation $o_t$ is defined by

$$\hat{r}(o_t) = \log(D_{rew}(BE_T(o_t))) - \log(1 - D_{rew}(BE_T(o_t))), \tag{28}$$

which is in a similar form to that in [24].

# B    Algorithm Pseudo Codes

---

**Algorithm 1** Dual feature extraction and image generation

---

**Input:** The number of epochs $n_{epoch\_it}$, domain encoders $DE = (DE_S, DE_T)$, behavior encoders $BE = (BE_S, BE_T)$, generators $G = (G_S, G_T)$, feature discriminators $FD = (BD_B, DD_B, BD_D, DD_D)$, image discriminators $ID = (ID_S, ID_T)$, observation sets $O_{SE}, O_{SN}, O_{TN}$ of size $n_{demo}$.

  Initialize parameters for $DE, BE, G, FD, ID$.
  Make a copy of $O_{TN}$ and initialize $O_{TL}$ with the copy.
  **for** $k = 1$ to $n_{epoch\_it}$ **do**
    Sample a minibatch of observations $o_{SE}, o_{SN}, o_{TN}, o_{TL}$ from $O_{SE}, O_{SN}, O_{TN}, O_{TL}$, respectively.
    **for** $o$ in $(o_{SE} \cup o_{SN} \cup o_{TN} \cup o_{TL})$ **do**
      Extract $DE(o)$ and $BE(o)$.
      Generate $\hat{o}$ in eq. (14).
      Extract $DE(\hat{o})$ and $BE(\hat{o})$.
    **end for**
    **for** $o_x, o_y$ (not in the same domain) in $(o_{SE} \cup o_{SN} \cup o_{TN} \cup o_{TL})$ **do**
      Generate $\hat{o}_{(x,y)}$ and $\hat{o}_{(y,x)}$ in eqs. (18) and (19).
      Generate $G(DE(\hat{o}_{(x,y)}).BE(\hat{o}_{(y,x)}))$.
    **end for**
    Compute $L_{E,G}$ and $L_{FD,ID}$ in eqs. (25) and (26).
    Update $DE, BE$ and $G$ to minimize $L_{E,G}$.
    Update $FD$ and $ID$ to minimize $L_{FD,ID}$.
  **end for**
  **return** $E, G, FD, ID$

---

---

**Algorithm 2** Reward estimation and policy update

---

**Input:** The number of training epochs $n_{epoch\_pol}$, the number of discriminator updates $n_{update,D}$, the number of policy updates $n_{update,\theta}$, domain encoders $DE$, behavior encoders $BE$, generators $G$, discriminator $D_{rew}$ for reward estimation, learner policy $\pi_\theta$, observations sets $O_{SE}, O_{SN}, O_{TN}$ of size $n_{demo}$, replay buffer $B$ of size $n_{buffer}$.

  Initialize parameters for $D_{rew}, \pi_\theta$ and $B$.
  **for** $k_1 = 1$ to $n_{epoch\_pol}$ **do**
    Sample a trajectory $\tau \sim \pi_\theta$.
    Store transitions $(s, a, s')$ and observations $o$ to $B$.
    **for** $k_2 = 1$ to $n_{update,D}$ **do**
      Sample minibatch of observations $o_{SE} \in O_{SE}, o_{TN} \in O_{TN}, o_{TL} \in B$.
      Update $D_{rew}$ to minimize $L_D$ in Sec. 5.4.
    **end for**
    **for** $k_3 = 1$ to $n_{update,\theta}$ **do**
      Sample minibatch of transitions $(s, a, s')$ and corresponding observations $o$.
      Compute reward $\hat{r}$ in Sec. 5.4.
      Update the policy $\pi_\theta$ using SAC.
    **end for**
  **end for**

---

# C   Third-Person Imitation Learning (TPIL)

This section summarises TPIL [37], one of the pioneering works that address the domain shift problem in IL. TPIL trains a model based on an unsupervised domain adaptation technique [15] with GAIL [18]. TPIL consists of a single behavior encoder $BE$, a domain discriminator $DD$, and a behavior discriminator $BD$, as shown in Fig. 12. The input label $XB_X$ in Fig. 12 means that the input is in the $X$ domain with behavior $B_X$. $X$ can be source $S$ or target $T$, and $B_X$ can be expert $E$ or non-expert $N$. The key idea is to train $BE$ to extract domain-independent behavior features from inputs. The trained $BE$ can be used to tell whether the learner's action in the target domain is expert behavior or non-expert behavior, and we can use this evaluation to train the learner. $DD$ learns to predict the domain label (source or target) of the input, and $BD$ learns to predict the behavior label (expert or non-expert) of the input. Therefore, encoder $BE$ learns to fool $DD$ by removing domain information from the input while helping $BD$ by preserving behavior information from the input. In [37], the behavior feature is a concatenation of $BE(o_t)$ and $BE(o_{t-4})$; however, we just simply the notation and denote the behavior feature by $BE(o_t)$.

The total loss for TPIL $\mathcal{L}_{TPIL}$ is defined as follows:

$$\mathcal{L}_{TPIL} = \sum_{o_i} \mathcal{L}_{CE}(BD(BE(o_i)), b_i) + \lambda_d \mathcal{L}_{CE}(DD(\mathcal{G}(BE(o_i)), d_i)) \tag{29}$$

where $o_i$ is an image observation, $\mathcal{L}_{CE}$ is the cross-entropy loss, $d_i$ is the domain label of $o_i$ (1 for source domain and 0 for target domain), $b_i$ is the behavior label of $o_i$ (1 for expert behavior and 0 for non-expert behavior), $\lambda_d$ is a hyperparameter, and $\mathcal{G}$ is a Gradient Reversal Layer (GRL) [15] is defined by

$$\mathcal{G}(x) = x$$
$$d\mathcal{G}(x)/dx = -\lambda_g I$$

where $\lambda_g$ is a hyperparameter. GRL enables updating $BE$, $DD$, and $BD$ simultaneously using back-propagation. The imitation reward $r_t$ for an observation $o_t$ generated by the learner policy is defined by the probability that $BD$ predicts the observation to be generated by an expert policy, as determined by $r_t = BD(BE(o_t))$.

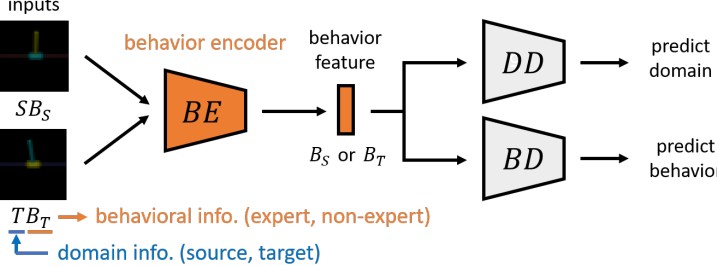

Figure 12: Basic structure for domain-independent behavior feature extraction: $BE$ - behavior encoder, $BD$ - behavior discriminator, $DD$ - domain discriminator

# D Implementation Details

## D.1 Network Architectures

In this section, we explain the network architectures of the components of the proposed method. Table 4 shows the layers for encoders, generators, and discriminators.

Each behavior encoder ($BE_S$ or $BE_T$) consists of 6 convolutional layers and a flattened layer. The number of output channels for each convolutional layer is (16, 16, 32, 32, 64, 64), and each channel has $3 \times 3$ size. Zero padding is applied before each convolutional layer, and ReLU activation is applied after each convolutional layer except the last layer. The flattening layer reshapes the input to a one-dimensional vector. The shape of the input is $(N, W, H, 4C)$, where $N$ is the minibatch size, $W$ and $H$ are the width and height of each image frame, and $C$ is the number of channels (for RGB images, $C = 3$). The shape of the behavior feature vector is $(N, (W/4) \times (H/4) \times 64)$.

Each domain encoder ($DE_S$, $DE_T$) consists of 6 convolutional layers, a flattened layer, and a linear layer. The number of output channels for each convolutional layer is (16, 16, 32, 32, 64, 64), and each channel has $3 \times 3$ size. Zero padding is applied before each convolutional layer, and ReLU activation is applied after each convolutional layer. The shape of the input is $(N, W, H, C)$, and the shape of the domain feature vector is $(N, 8)$.

Each generator ($G_S$ or $G_T$) consists of 7 transposed convolutional layers. The number of output channels for each layer is (64, 64, 32, 32, 16, 16, $4C$), and each channel has $3 \times 3$ size. Zero padding is applied before each layer, and ReLU activation is applied after each layer except the last layer. The input is a behavior feature vector and a domain feature vector. The shape of the output is $(N, W, H, 4C)$, which is the same as the shape of the input of behavior encoders.

Each feature discriminator ($BD_B$, $DD_B$, $BD_D$, or $DD_D$) consists of 2 fully connected layers and a linear layer. Each fully connected layer has 32 hidden units with ReLU activation. Each image discriminator ($ID_S$ or $ID_T$) consists of 6 convolutional layers, a flattened layer, and a linear layer. Each convolutional layer has $3 \times 3$ channel size. The number of output channels for each layer is (16, 16, 32, 32, 64, 64). Zero padding is applied before each layer, and ReLU activation is applied after each layer except on the last layer. The discriminator $D$ for reward estimation consists of 2 fully connected layers and a linear layer. Each fully connected layer has 100 hidden units with ReLU activation. The input size is $(N, (W/4) \times (H/4) \times 64)$.

For the SAC algorithm, each actor and the critic consist of 2 fully connected layers and a linear layer. Each fully connected layer has 256 hidden units with ReLU activation.

## D.2 Details for Training Process

Before training the model, we collected expert demonstrations $O_{SE}$, and non-expert datasets $O_{SN}$, $O_{TN}$. $O_{SE}$ is obtained by the expert policy $\pi_E$, which is trained for 1 million timesteps in the source domain using SAC. $O_{SN}$ is obtained by a policy taking uniformly random actions in the source domain, and $O_{TN}$ is obtained by a policy taking uniformly random actions in the target domain. For each IL task, the number of observations (i.e., the number of timesteps) for $O_{SE}$, $O_{SN}$, $O_{TN}$ is $n_{demo} = 10000$ except for HalfCheetah-to-locked-legs task, where $n_{demo} = 20000$ for this task. Each observation consists of 4 RGB images, but the size of these images varies depending on the specific IL task. For IL tasks including IP, IDP, CartPole, and Pendulum, the image size is 32x32. For IL tasks including RE2 and RE3, the image size is 48x48. For IL tasks including HalfCheetah and UMaze, the image size is 64x64. Note that the proposed method does not require a specific input image size.

Our proposed method has two training phases. The first phase updates domain encoders, behavior encoders, generators, feature discriminators, and image discriminators. The second phase updates the discriminator $D$ for reward estimation and the policy $\pi_\theta$. In the first phase, we trained the model for $n_{epoch\_it} = 50000$ epochs for all IL tasks except for HalfCheetah-to-locked-legs task, where $n_{epoch\_it} = 200000$ for this task. In each epoch, we sampled a minibatch of size 8 from each dataset $O_{SE}, O_{SN}, O_{TN}, O_{TL}$. We set $O_{TL}$ as a copy of $O_{TN}$. For coefficients in Eqs. (25) and (26), we set $\lambda_{feat}^{pred} = \lambda_{feat}^{adv} = 0.01$, $\lambda_{feat}^{reg} = 0.1$, $\lambda_{img}^{adv} = 1$, $\lambda_{feat}^{sim} = \lambda_{feat}^{recon} = 1000$, $\lambda_{img}^{recon} = \lambda_{img}^{cycle} = 100000$. We used $\lambda_{feat}^{cycle} = 10$ for IP-to-color, IDP-to-one, IP-to-two, and PointUMaze-to-ant, $\lambda_{feat}^{cycle} = 100$

Table 4: Layers for networks in the proposed model. Conv(nc, st, act) denotes a 2D convolutional layer with the number of output channels (nc), stride (st), and activation (act). Conv(nc, st, act) denotes a 2D transposed convolutional layer with the number of output channels (nc), stride (st), and activation (act). FC(nh, act) denotes a fully connected layer with the number of units (nh) and activation (act). Flatten denotes a function that reshapes the input to a one-dimensional vector.

| $BE_S, BE_T$ | Conv(16, 1, ReLU) | $DE_S, DE_T$ | Conv(16, 1, ReLU) |
|---|---|---|---|
| | Conv(16, 1, ReLU) | | Conv(16, 1, ReLU) |
| | Conv(32, 2, ReLU) | | Conv(32, 2, ReLU) |
| | Conv(32, 1, ReLU) | | Conv(32, 1, ReLU) |
| | Conv(64, 2, ReLU) | | Conv(64, 2, ReLU) |
| | Conv(64, 1, Linear) | | Conv(64, 1, ReLU) |
| | Flatten | | Flatten |
| | | | FC(8, Linear) |
| $BD_B, DD_B$ | FC(32, ReLU) | $BD_D, DD_D$ | FC(32, ReLU) |
| | FC(32, ReLU) | | FC(32, ReLU) |
| | FC(1, Linear) | | FC(1, Linear) |
| $G_S, G_T$ | ConvTranspose(64, 1, ReLU) | $ID_S, ID_T$ | Conv(16, 1, ReLU) |
| | ConvTranspose(64, 1, ReLU) | | Conv(16, 1, ReLU) |
| | ConvTranspose(32, 2, ReLU) | | Conv(32, 2, ReLU) |
| | ConvTranspose(32, 1, ReLU) | | Conv(32, 1, ReLU) |
| | ConvTranspose(16, 2, ReLU) | | Conv(64, 2, ReLU) |
| | ConvTranspose(16, 1, ReLU) | | Conv(64, 1, ReLU) |
| | ConvTranspose($4C$, 1, Linear) | | Flatten |
| | | | FC(1, Linear) |
| $D$ | FC(100, ReLU) | | |
| | FC(100, ReLU) | | |
| | FC(1, Linear) | | |

for IDP-to-color, $\lambda_{feat}^{cycle} = 100$ for RE3-to-tilted, and $\lambda_{feat}^{cycle} = 10000$ for RE2-to-tilted, RE3-to-two, RE2-to-three, and HC-to-LF. We used $\lambda_{feat}^{cycle} = 100$ for all environments from DeepMind Control Suite. We chose these coefficients so that the scale of each loss component is balanced, which was observed to be enough to yield good performances of $\pi_\theta$.

As mentioned in Appendix A.4, we adjusted the values of $c_{norm,d}$ and $c_{norm,b}$ in Eq. (23) of the proposed method for the experiment. For IL tasks including IP, IDP, RE2, RE3, and HalfCheetah, we set $c_{norm,d} = c_{norm,b} = 0$. For IL tasks including CartPole and Pendulum, we set $c_{norm,d} = 1$ and $c_{norm,b} = 20$. For UMaze task, we set $c_{norm,d} = 1$ and $c_{norm,b} = 40$. The ratio between $c_{norm,d}$ and $c_{norm,b}$ is determined based on the input image size since the proposed method does not require a specific input image size and produces behavior features of varying sizes depending on the input image size.

In the second phase, we trained $D$ for reward estimation and $\pi_\theta$ for $n_{epoch\_pol}$ epochs, where $n_{epoch\_pol} = 20$ for IP-to-color, IDP-to-one tasks, $n_{epoch\_pol} = 30$ for CartPole and Pendulum tasks in DeepMind Control Suite, and $n_{epoch\_pol} = 100$ for other IL tasks. Each epoch consists of 10000 timesteps. In each epoch, the following process is repeated. We sampled a trajectory from $\pi_\theta$, and stored state transitions $(s, a, s')$ and observations $o$ to the replay buffer $B$. The maximum size of $B$ is $n_{buffer} = 100000$, which is much smaller compared to that in off-policy RL algorithms because of the large memory consumption when storing images in $B$. Then, $D$ is updated for $n_{update,D}$ times. We $n_{update,D}$ to be 50 times smaller than the episode length for each IL task. For example, if the episode length is 1000, then $n_{update,D} = 20$. For every $D$ update, we sampled a minibatch of size 128 from each dataset $O_{SE}$ and $O_{TN}$. For $O_{TL}$, we sampled a minibatch of size 64 from $B$ and a minibatch of size 64 from $O_{TN}$. After $D$ update, $\pi_\theta$ is updated for $n_{update,\theta}$ times using SAC [17]. We set $n_{update,\theta}$ to be equal to the episode length for each IL task. For example, if the episode length is 1000, then $n_{update,\theta} = 1000$. For every $\pi_\theta$ update, we sampled a minibatch of size 256 from $B$.

In each epoch, we evaluated $\pi_\theta$ after updating $D$ and $\pi_\theta$. We sampled $n_{eval}$ trajectories using $\pi_\theta$ and computed the average return, where each return is computed based on the true reward in the target domain. We set $n_{eval} = 10$ as default. For IL tasks including Reacher environments, we set

$n_{eval} = 200$ to reduce variance because each return highly depends on the goal position in Reacher environments. We used Adam optimizer [23] for optimizing all networks. We set the learning rate $lr = 0.001$ and momentum parameters $\beta_1 = 0.9$ and $\beta_2 = 0.999$ for encoders, generators, and discriminators. We set $lr = 0.0003$, $\beta_1 = 0.9$ and $\beta_2 = 0.999$ for the actor and critic. For SAC, we set the discount factor $\gamma = 0.99$, the parameter $\rho = 0.995$ for Polyak averaging the Q-network, and the entropy coefficient $\alpha = 0.1$.

The proposed method is implemented on TensorFlow 2.0 with CUDA 10.0, and we used two Intel Xeon CPUs and a TITAN Xp GPU as the main computing resources. One can use a GeForce RTX 3090 GPU instead as the computing resource, which requires TensorFlow 2.5 and CUDA 11.4. Based on 32x32 images, The GPU memory consumption is about 2.5GB for the training feature extraction model and about 1.5GB during policy update. Note that these highly depend on the image size and the batch size. Our D3IL is implemented based on the code provided by the authors of [6].

### D.3 Dataset and Training Process for PointUMaze-to-Ant Task

For the UMaze environments, the number of observations for $O_{SE}, O_{SN}$ or $O_{TN}$ is $n_{demo} = 10000$. Each observation consists of 4 frames and each frame is a 64x64 RGB image. During the training, the reward that the agent receives for each timestep is $r_{total} = c \times (r_{IL} + \mathbf{1}_{reach\_goal} \times r_{goal})$, where $r_{IL}$ is the estimated reward given by $D$; $\mathbf{1}_{reach\_goal}$ is 1 if the robot reaches the goal and 0 otherwise; $r_{goal}$ is the reward when the robot reaches the goal; and $c > 0$ is a scaling coefficient. In the experiment, we chose $c = 100$ and $r_{goal} = 1$, and these parameters are applied to all baselines. For the proposed method, the number of epochs, minibatch sizes, and network structures for the training feature extraction network and image reconstruction network are the same as those in Appendix D.1. The policy is trained for $n_{epoch\_pol} = 200$ and each epoch consists of 10000 timesteps, so the total number of timesteps for policy training is 2,000,000.

Also, we trained an agent using a vanilla RL method (SAC [17]) with access to true rewards, to show that it is difficult to obtain expert demonstrations in the target domain. For SAC, the agent receives true rewards. For true rewards, the agent receives a penalty of -0.0001 for each timestep and receives 1000 when the robot reaches the goal, so the maximum true episodic return is 1000.

# E   RL Environment Settings and Sample Image Observations

We first describe the base RL tasks in the Gym.

*Inverted pendulum (IP):* The agent moves a cart to prevent the pole from falling. The episode length is 1000. Unlike the IP task in Gym [4], an episode does *not* terminate midway even if the pole falls. This makes the task much more challenging because each episode provides less useful observations to solve the task.

*Inverted double pendulum (IDP):* This is a harder version of the IP task where there are two poles on the cart.

*Reacher-two (RE2):* The agent moves a two-link arm with one end fixed so that the movable endpoint of the arm reaches the target point. The episode length is 50. Unlike the Reacher task in Gym, we narrowed the goal point candidates to 16 points, where the polar coordinate $(r, \varphi)$ of the target point can be $r \in \{0.15, 0.2\}$, and $\varphi \in \{0, \pi/4, \pi/2, 3\pi/4, \cdots, 7\pi/4\}$.

*Reacher-three (RE3):* This is a harder version of the RE2 task where the agent moves a three-link armed robot.

Note that our IL tasks are more difficult than those considered in the experiments in the baseline algorithm references [6, 37]. Our simulation setting has an episode length of 1000 for the IP and IDP tasks instead of 50. The episode horizon is also a component of an MDP, and the observation space of the MDP changes as the episode horizon changes. MDP with a longer episode length can yield more various observations which are much different from the initial one, so the observation gap between the source domain and the target domain increases. We meant this for a bigger domain gap.

If the goal position is fixed throughout the entire learning phase in RE tasks, as designed in [6, 37], distinguishing between expert and non-expert behavior solely depends on the arm position. However, in our implementation, the goal position is randomly chosen from 16 positions for each episode. This makes distinguishing expert and non-expert behavior much more challenging because the same arm movement can be considered expert or non-expert behavior depending on the goal position. Therefore, extracting proper behavior features is much more difficult as the feature extraction model should catch the relationship between the arm movement and the goal position.

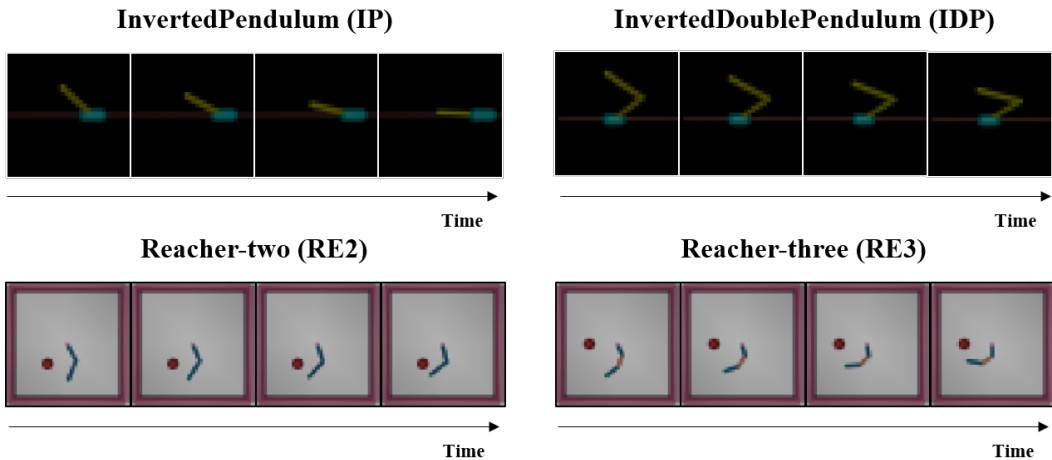

Figure 13: Example observations for IP, IDP, RE2, and RE3 environment

The following images are sample observations for IL tasks with changing a robot's degree of freedom.

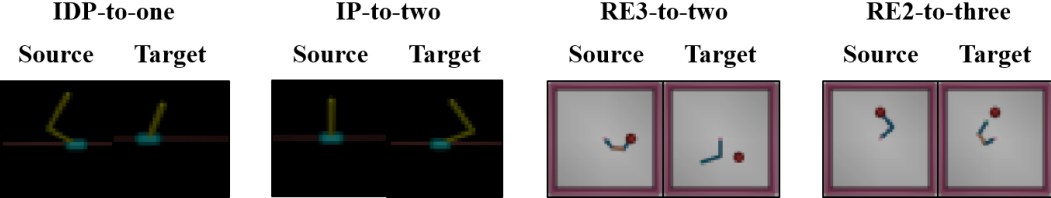

Figure 14: Example image pairs for IL tasks with changing DOF

In the HalfCheetah (HC) environment, the agent controls an animal-like 2D robot. The goal is to move the animal-like 2D robot forward as fast as possible by controlling 6 joints. Similarly to [6], we have a modification with immobilized feet, and this modified environment is 'HalfCheetah-Locked-Feet (HC-LF)'. In the HC-LF environment, the robot can use only 4 joints instead of 6, where the immobilized feet are colored in red.

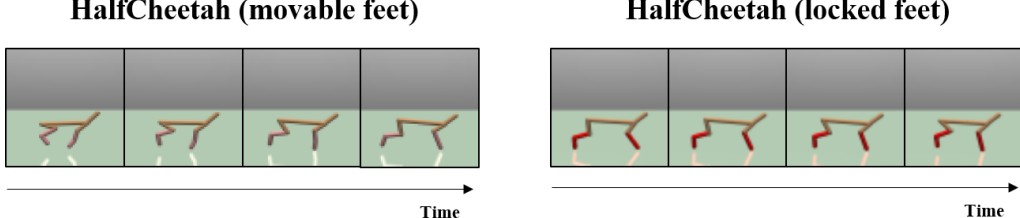

Figure 15: Example observations of HalfCheetah environment

The following images are sample image observations of environments in the DeepMind Control Suite (DMCS). We use CartPole-Balance, CartPole-SwingUp, and Pendulum environments from DMCS. The goal for all environments is to keep the pole upright.

*CartPole-Balance*: The pole starts in an upright position and the agent should move the cart to keep the pole upright while avoiding exceeding the boundaries along the x-axis.

*CartPole-Swingup*: The pole starts in a downward position and the agent should move the cart to swing the pole and keep the pole in an upright position while avoiding exceeding the boundaries along the x-axis.

*Pendulum*: The agent should add torque on the center to swing the pole and keep the pole upright.

Domain adaptation between these tasks is challenging because not only the embodiment between agents in both domains are different but also the distributions of expert and non-expert demonstrations, and initial observations for both domains are quite different.

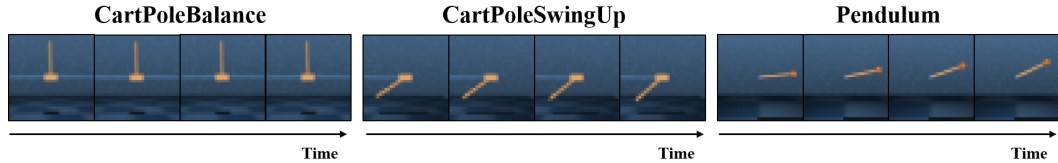

Figure 16: Example observations of environments in DMCS

We omit sample image observations of PointUMaze and AntUMaze environments because they are already provided in the main paper. Table 5 shows the dimensions of state space, action space, and visual observation for RL tasks used in the Experiment section.

Table 5: State dimension, action dimension, and visual observation image size for RL tasks used in the Experiments section. Each visual observation input consists of 4 RGB images, and the rightmost column shows the size (width x height) of each image.

| RL Task | State dimension | Action dimension | Visual observation image size |
|---|---|---|---|
| Inverted Pendulum (IP) | 4 | 1 | 32x32 |
| Inverted Double Pendulum (IDP) | 11 | 1 | 32x32 |
| Reacher-two (RE2) | 11 | 2 | 48x48 |
| Reacher-three (RE3) | 14 | 3 | 48x48 |
| HalfCheetah (HC) | 17 | 6 | 64x64 |
| HalfCheetah-Locked-Feet (HC-LF) | 13 | 4 | 64x64 |
| Cartpole | 5 | 1 | 32x32 |
| Pendulum | 3 | 1 | 32x32 |
| PointUMaze | 7 | 2 | 64x64 |
| AntUMaze | 30 | 8 | 64x64 |

# F    Additional Experimental Results

## F.1    Results on IL Tasks with Changing Visual Effect

We evaluated D3IL on tasks where the domain difference is caused by visual effects on image observations on four IL tasks: IP and IDP, with different color combinations of the pole and cart (IP-to-colored and IDP-to-colored tasks), and RE2 and RE3, with different camera angles (RE2-to-tilted and RE3-to-tilted tasks). Sample image observations are provided in Figure 17. For the IP and IDP tasks, the primary changes occur in the pixel values of the pole and the cart. Meanwhile, for the RE2 and RE3 tasks, major changes are observed in the pixel values of the background outside of the arms and the goal position. The average episodic return for D3IL and the baselines over 5 seeds are shown in Fig. 18. We only included baselines that receive demonstrations as images because changing visual effects on image observations do not affect the true state space and the action space. As shown in Fig. 18, D3IL outperformed the baselines with large margins on IL tasks with changing visual effects on image observations.

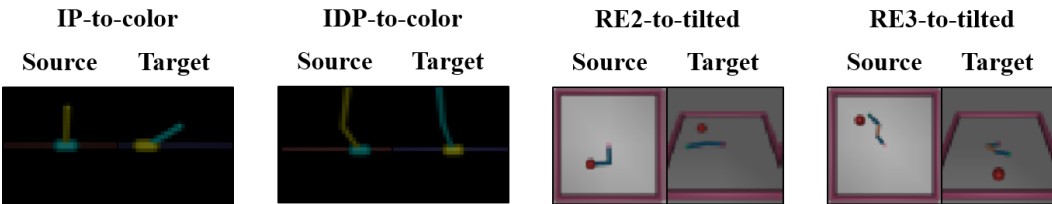

Figure 17: Example image pairs for IL tasks with changing visual effects

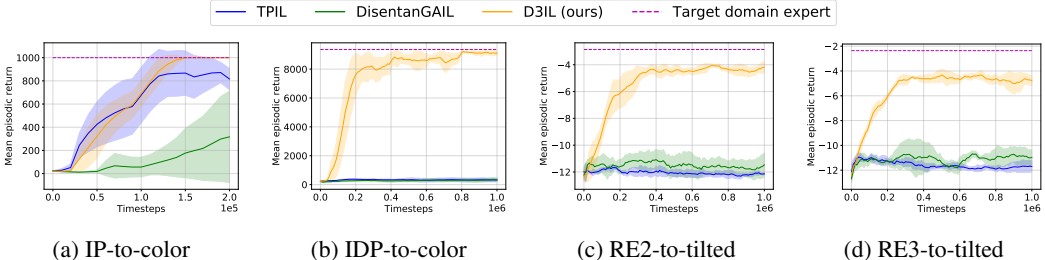

(a) IP-to-color         (b) IDP-to-color         (c) RE2-to-tilted         (d) RE3-to-tilted

Figure 18: The learning curves in the target domain of the IL tasks with visual effects

## F.2 The Evaluation on the Feature Quality

Fig. 19 shows some instances of the estimated rewards from sampled observations in the RE2-to-three task, corresponding to the time of 150,000 timesteps in Fig. 5d. The left column of Fig. 19 is non-expert behavior and the right column is expert behavior. One side of the Reacher arm is fixed at the center of the image. In the left column, the movable arm end moves away from the goal position through four time-consecutive images, whereas the movable arm end moves towards the goal position in the right column. Note that in these instances, the baseline algorithm assigns a higher reward to the non-expert behavior, whereas the proposed method correctly assigns a higher reward to the expert behavior. We conjecture that this wrong reward estimation of the baseline algorithms based on poor features is the cause of the gradual performance degradation of the baseline algorithms in Fig. 5.

In Fig. 20, we checked the quality of extracted features from a conventional method, TPIL, and that of our method. Since TPIL does not consider image reconstruction but considers only the extraction of domain-independent behavior features. In order to generate an image from the domain-independent behavior feature of TPIL, along with TPIL we also trained two generators $(G_S, G_T)$: $G_S$ takes a behavior feature vector as input and generates an image that resembles the images from the source domain. $G_T$ takes a behavior feature vector as input and generates an image that resembles the images from the target domain. This $G_S$ and $G_T$ training is done with two discriminators ($ID_S$ and $ID_T$), where $ID_S$ distinguishes the real and generated images in the source domain, and $ID_T$ does so in the target domain. As shown in Fig. 20, the reconstructed image from the behavior feature of TPIL is severely broken for the RE2-to-tilted task. Thus, it shows that the "domain-independent" behavior feature of TPIL does not preserve all necessary information of the original observation image. On the other hand, our method based on dual feature extraction and image reconstruction, yields image reconstruction looking like ones. This implies that the features extracted from our method well preserve the necessary information, and thus reconstruction is well done.

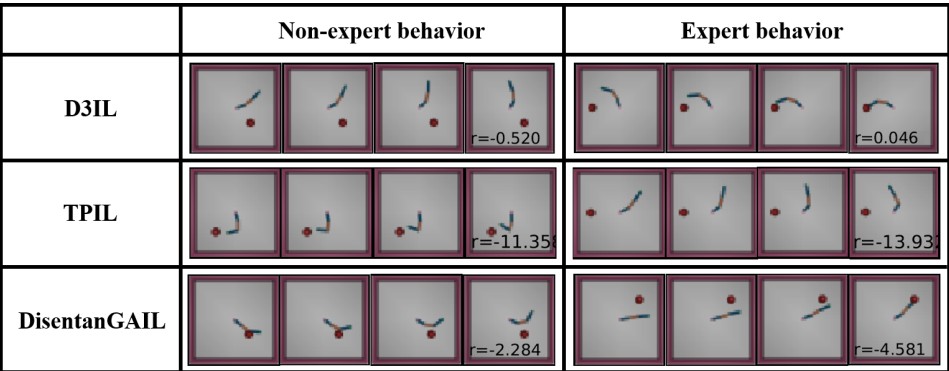

Figure 19: RE2-to-three: Instances of estimated rewards. Four time-consecutive images comprise one observation. The estimated reward is shown in the fourth image.

| | Conventional IL (TPIL) | | | | D3IL | | | |
|---|---|---|---|---|---|---|---|---|
| **IL task** | **IP-to-two** | | **RE2-to-tilted** | | **IP-to-two** | | **RE2-to-tilted** | |
| **Domain** | Source | Target | Source | Target | Source | Target | Source | Target |
| **True images observations** | | | | | | | | |
| **Imaged generated to a different domain** | | | | | | | | |

Figure 20: Examples of real and generated images. The generated images are generated from the feature vectors obtained by conventional TPIL and our proposed method. Here we used IP-to-two and RE2-to-three task, where each task is explained in Sec. 6.1.

### F.3 Experimental Results on Simpler Tasks

In the main result subsection of Sec. 6, we mentioned that the baseline algorithms perform well in the case of easier settings. As a preliminary experiment, we verify the performance of the proposed method and the baseline IL algorithms on simpler IL tasks, which are easier variants of the IL tasks shown in Sections 6.2 and 6.3. We first describe the base RL tasks.

*Inverted pendulum - short (IPshort)*: This is the same task as the IP task, but the episode length is 50.

*Inverted double pendulum - short (IDPshort)*: This is the same task as the IDP task, but the episode length is 50.

*Reacher-two - one goal (RE2onegoal)*: This is the same task as the RE2 task, but the goal point is fixed to $(x, y) = (1.0, 1.0)$ for the entire training.

*Reacher-three - one goal (RE3onegoal)*: This is the same task as the RE3 task, but the goal point is fixed to $(x, y) = (1.0, 1.0)$ for the entire training.

Sample visual observations for each IL task are provided in Fig. 21. We followed the training process as described in [6]. For each IL task, we collected $O_{SE}$, $O_{SN}$ and $O_{TN}$ of size 10000. The learner policy $\pi_\theta$ is trained for 20 epochs. Each epoch has 1000 timesteps, so the total number of timesteps is 20000. The size of the replay buffer is 10000. We used TPIL [37] and DisentanGAIL [6] as baseline algorithms. For each IL task, we trained the learner policy using our proposed method and baseline algorithms over 5 seeds. For evaluation, we generated 5 trajectories from $\pi_\theta$ for every epoch and computed the average return, where the return was computed based on the true reward in the target domain.

Fig. 22 shows the average episodic return. As shown in Fig. 22, the baseline algorithms perform well in these easy IL tasks. The IL tasks shown in Fig. 21 are much easier to solve than the IL tasks in Sections 6.2 and 6.3 because the pole needs to be straight up for only 50 timesteps instead of 1000 timesteps in IPshort and IDPshort, and the goal position is fixed to a single point instead of randomly sampled from 16 candidates in RE2ongoal and RE3onegoal. The baselines can achieve good performance on these easy IL tasks. However, as shown in Fig. 5 of the main paper and Fig. 18 in Appendix 6.2, the baseline algorithm yields poor performance on hard IL tasks and the proposed method achieves superior performance on hard IL tasks to the baselines due to the high-quality feature extraction by the dual feature extraction and image reconstruction.

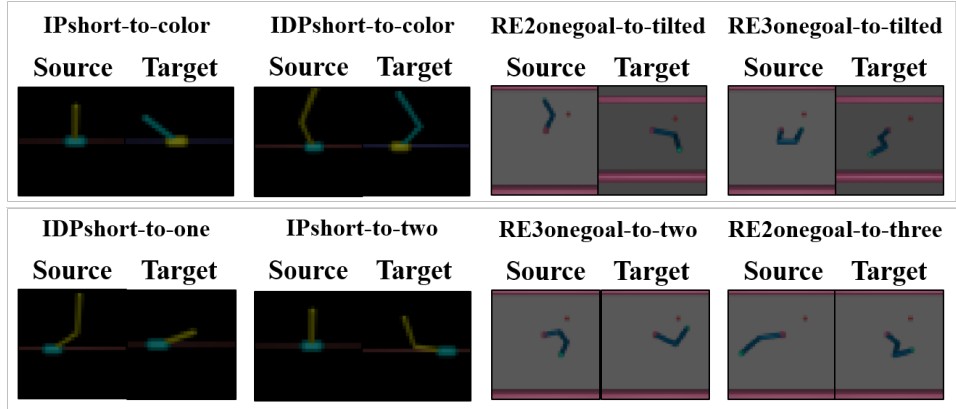

Figure 21: Example observation pairs for each IL task. For each pair of images, the left-hand side is the image in the source domain, and the right-hand side is the image in the target domain.

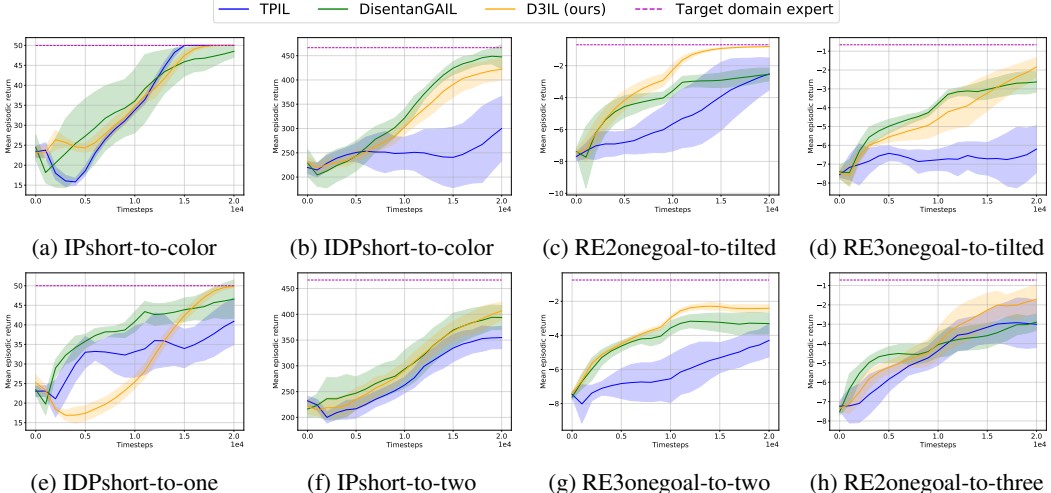

(a) IPshort-to-color   (b) IDPshort-to-color   (c) RE2onegoal-to-tilted   (d) RE3onegoal-to-tilted

(e) IDPshort-to-one   (f) IPshort-to-two   (g) RE3onegoal-to-two   (h) RE2onegoal-to-three

Figure 22: The learning curves for the simpler IL tasks in the target domain

### F.4 Experiments Results with Single Expert Trajectory during Policy Update

In this section, we investigated whether D3IL can successfully train the target learner when a single trajectory of expert demonstration is available during policy training. For results in Section 6, multiple trajectories of expert demonstrations and non-expert data are provided for D3IL, TPIL, and DisentanGAIL, where the details are provided in Appendix D. Since GWIL calculates the Gromov-Wasserstein distance between expert and learner policies, GWIL can leverage only a single expert trajectory. This is also true for the implementation with the code provided by the authors of [9]. In contrast, the size of the replay buffer $n_{buffer}$ of GWIL equals the number of training timesteps, which can vary from 200,000 to 2,000,000 depending on the IL task. On the other hand, $n_{buffer}$ is fixed to 100,000 for D3IL and baselines that utilize image observations. This is due to the huge memory consumption for storing image observations. In this experiment, we evaluated the performance of D3IL under the condition of a single trajectory of expert demonstration during policy training. To provide one trajectory of $SE$, and similarly, we supplied $SN$ and $TN$ with the same size of $SE$. In Reacher environments, we collected one trajectory per goal position for each $SE$, $SN$, and $TN$, respectively. Fig. 23 shows the average return of D3IL and baselines across 5 seeds.

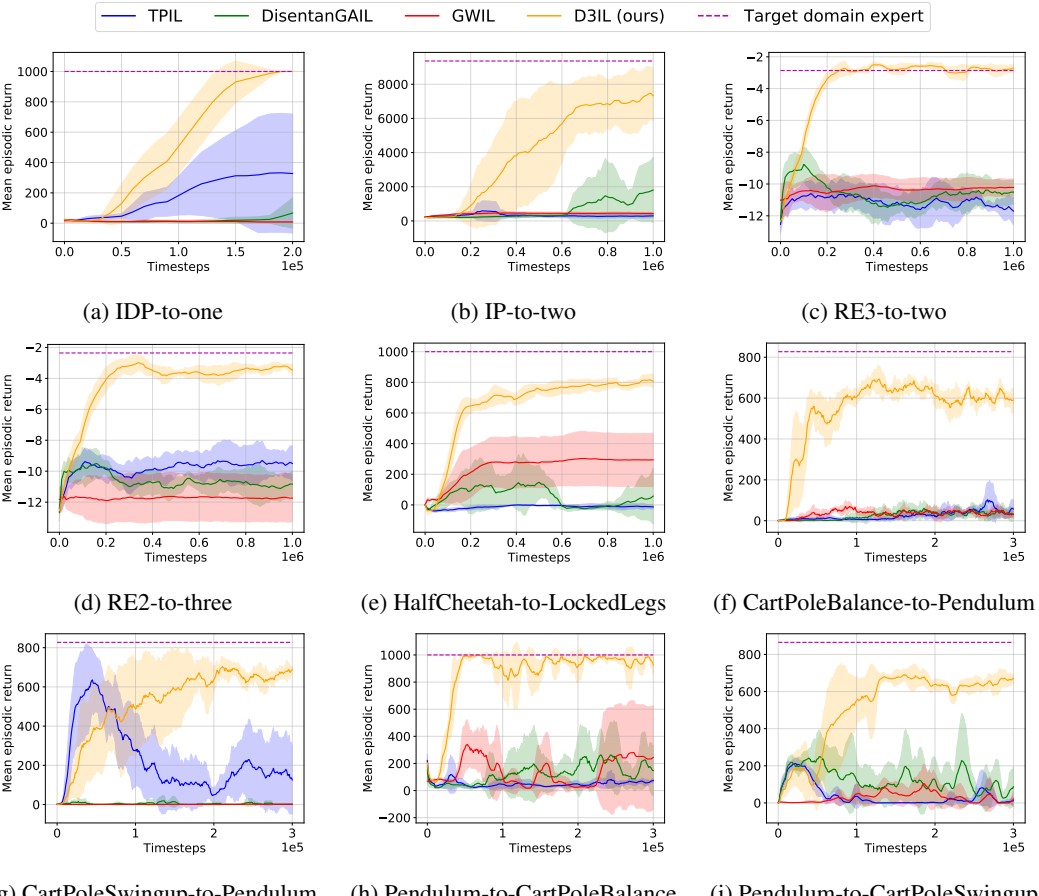

Figure 23: The learning curves for the considered IL tasks when using single expert trajectory

## F.5 Domain Transfer between CartPole-Balance and CartPole-SwingUp

In Section 6.5, we present the results of four IL tasks that use three different environments in DMCS: CartPole-Balance, CartPole-SwingUp, and Pendulum. In these tasks, the robot's embodiment and dynamics of both domains are the same but their task configuration and initial state distribution of both domains are different. Fig. 24 shows the results of the other two IL tasks where both domains use the CartPole environment. We evaluated D3IL and the baselines on IL tasks for 5 seeds. As illustrated in Fig. 24, all methods failed to solve the CartPole-SwingUp task using demonstrations from the CartPole-Balance task. In contrast, our methods were able to successfully solve the CartPole-Balance task using demonstrations from the CartPole-SwingUp task. This difference in performance can be attributed to the variation in expert demonstrations between the two environments. Demonstrations of the CartPole-SwingUp task typically show images in which the pole is upright, with only a small number of images where the pole is initially pointing downwards and begins to swing up. As a result, an agent for the CartPole-Balance task can learn that expert behavior involves keeping the pole upright and preventing it from falling. In contrast, demonstrations of the CartPole-Balance task typically only show images in which the pole is upright, which means that an agent designed for the CartPole-SwingUp task may not have access to information on expert behavior for swinging up the pole from a downward-pointing position. This problem can be mitigated if non-expert data contains more diverse images from the CartPole-SwingUp task. Our method can also be enhanced by utilizing images generated from the target learner to train our feature extraction model.

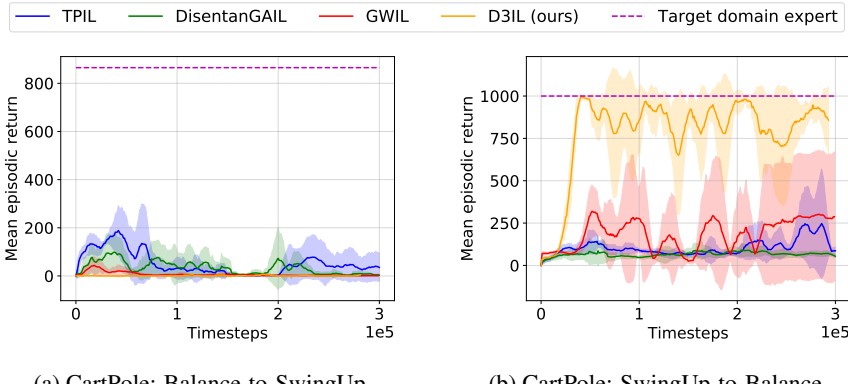

(a) CartPole: Balance-to-SwingUp          (b) CartPole: SwingUp-to-Balance

Figure 24: The learning curves for the considered IL tasks for CartPole-Pendulum and CartPole-SwingUp environments

### F.6 Results on Acrobot Environment

In this section, we evaluated D3IL on IL tasks where the target domain is the Acrobot environment in DMCS. In Acrobot, the agent should apply torque to the joint located *between* a two-link pole to make the pole stand upright. The dynamics of the Acrobot environment are much different from those of other environments like CartPole and Pendulum. Although the state and action dimensions of the Acrobot environment are low, achieving high scores in this task is non-trivial. In our implementation, the policy using vanilla RL with SAC, with 10 million timesteps, 256 hidden units for actor and critic networks, and a batch size of 256, obtained the average return of 45.49 across 5 seeds. Figure 25 shows sample observations for the Acrobot environment, where each observation consists of four 32x32 RGB images with corresponding consecutive timesteps.

We evaluated D3IL and baselines on two IL tasks, where the source domain is either CartPole-SwingUp or Pendulum, and the target domain is Acrobot. CartPole-SwingUp and Pendulum are known to be solvable by SAC. We examined whether D3IL can transfer expert knowledge from simpler tasks to more challenging ones. Fig. 26 shows the average episodic return of D3IL and the baselines across 5 seeds. The results demonstrate that D3IL outperforms other baselines, and achieves the average return of vanilla RL using SAC with 10 million timesteps in much smaller timesteps. Our method has the potential to improve if additional information on Acrobot's dynamics is provided to the model. This can be accomplished by obtaining diverse non-expert data, such as observations from the policy with medium quality or observations from the target learner policy, to train the feature extraction model.

## Acrobot

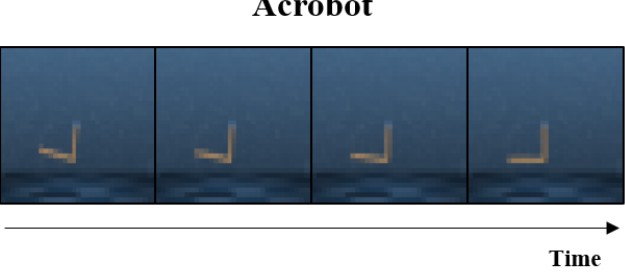

Figure 25: Sample observations on Acrobot task.

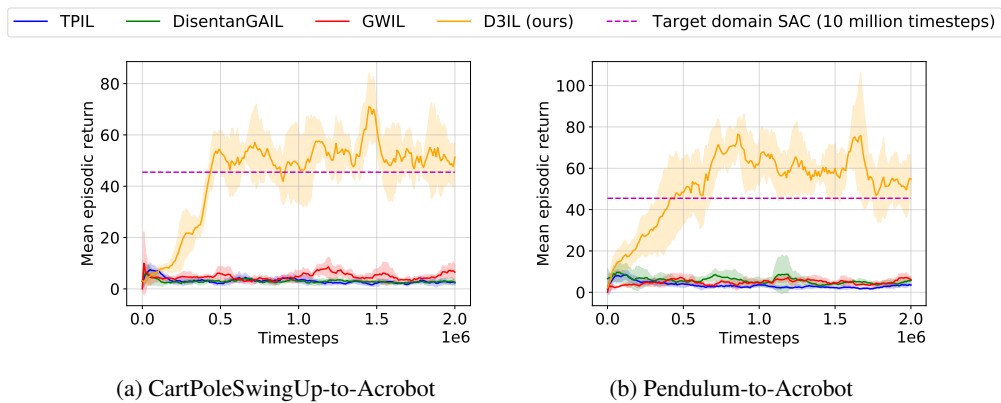

(a) CartPoleSwingUp-to-Acrobot       (b) Pendulum-to-Acrobot

Figure 26: The learning curves for the considered IL tasks with Acrobot as the target domain task

### F.7 Experiment on Walker, Cheetah, and Hopper Task

To assess the effectiveness of our proposed method in more challenging scenarios, we tested our method on IL tasks with domain shifts caused by variations in robot morphology. We employed Walker, Cheetah, and Hopper environments in the DeepMind Control Suite. In these environments, the objective is to propel a 2D robot forward as fast as possible by manipulating its joints. The difficulty arises from the significant difference in the robot morphology and dynamics between the source and target domains. Sample visual observations for each IL task are provided in Fig. 27.

For all environments, we set each episode length to be 200. The expert policy $\pi_E$ is trained for 3 million timesteps in the source domain using SAC. The number of observations is $n_{demo} = 1000$ (5 trajectories and each trajectory has a length of 200). Each observation consists of 4 frames and each frame is a 64x64 RGB image.

We evaluated our proposed method and baseline algorithm in three tasks: Walker-to-Cheetah, Walker-to-Hopper, and Cheetah-to-Hopper (denoted as 'A-to-B' where 'A' is the source domain and 'B' is the target domain). A camera generates image observations by observing the robot's movement. Rewards were assigned to the agent based on the robot's horizontal velocity, with higher velocity leading to greater rewards. Fig. 28 depicts the learning curve of our method and baselines on three tasks, where each curve shows the average horizontal velocity of the robot in the target domain over three runs. These results demonstrate the superiority of our method over the baselines, which leads us to anticipate that our proposed method can extend to more complicated domains.

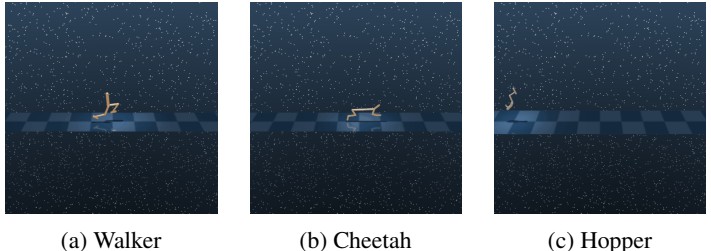

| (a) Walker | (b) Cheetah | (c) Hopper |
|:----------:|:-----------:|:----------:|

Figure 27: Sample observations for Walker, Cheetah, and Hopper task.

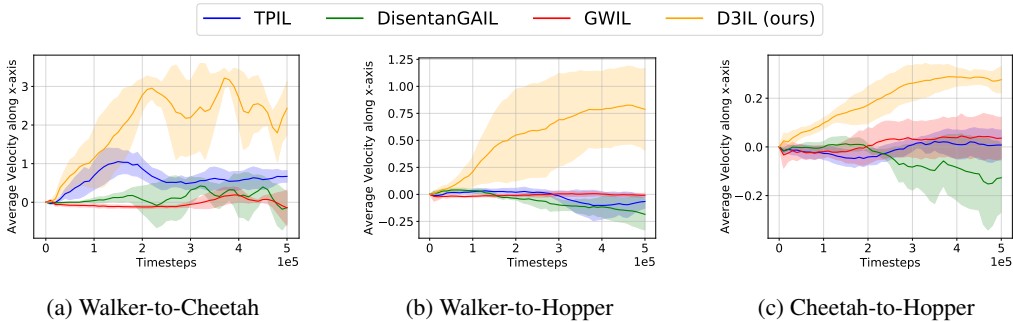

| (a) Walker-to-Cheetah | (b) Walker-to-Hopper | (c) Cheetah-to-Hopper |
|:---------------------:|:--------------------:|:---------------------:|

Figure 28: Results on Walker, Cheetah, and Hopper task

### F.8 Visualizing Sample Trajectories for Trained Learner

In this section, we provide visualized sample trajectories of demonstration sequences in the source domain and sequences generated by the learned policy trained in the target domain for qualitative analysis on two tasks: RE2-to-three and PointUMaze-to-Ant. Fig. 29 shows sample trajectories from the source domain expert and target domain learner on the RE2-to-three task. The top row of Fig. 29 shows a sample trajectory of image observations sampled by $\pi_E$ in the source domain using SAC for 1 million timesteps. The bottom row of Fig. 29 shows a sample trajectory of image observations sampled by $\pi_\theta$ trained using D3IL for 1 million timesteps in the target domain.

**Source domain: RE2 – Expert demonstration**

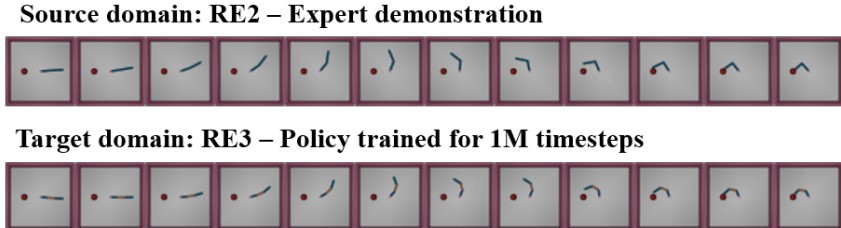

**Target domain: RE3 – Policy trained for 1M timesteps**

Figure 29: (Top) An example trajectory of image observations generated by $\pi_E$ in the source domain. (Bottom) An example trajectory of image observations generated by $\pi_\theta$ trained using D3IL for 1 million timesteps in the target domain.

In Section 6.6, we tested the performance of D3IL on the PointUMaze-to-Ant task. Fig. 30 shows sample trajectories from the source domain expert and target domain learner. The top row of Fig. 30 shows a sample trajectory of image observations sampled by $\pi_E$ in the source domain using SAC for 0.1 million timesteps. The bottom row of Fig. 30 shows a sample trajectory of image observations sampled by $\pi_\theta$ trained using D3IL for 2 million timesteps in the target domain. The episode lengths for the trained learner vary when running multiple evaluation trajectories, most of which have episode lengths in the range of 60 to 80. This implies that D3IL can train the agent in the target domain with its dynamics by observing expert demonstrations in the source domain with a different embodiment.

**Source domain: Point robot – Expert demonstration (episode length=18)**

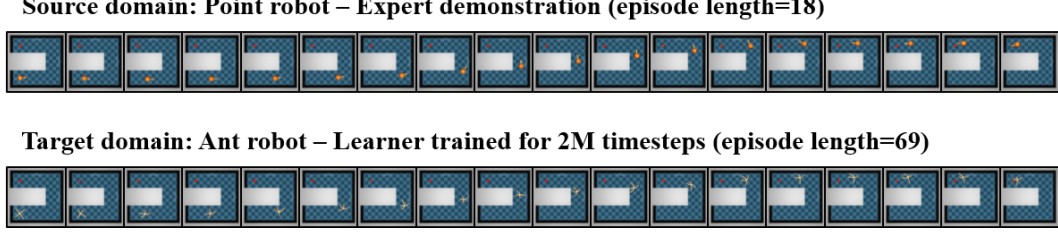

**Target domain: Ant robot – Learner trained for 2M timesteps (episode length=69)**

Figure 30: (Top) An example trajectory of image observations generated by $\pi_E$ in the source domain. In this figure, the interval between two consecutive image frames is 1 timestep, and the total episode length is 18. (Bottom) An example trajectory of image observations generated by $\pi_\theta$ trained using D3IL for 2 million timesteps in the target domain. In this figure, the interval between two consecutive image frames is 4 timesteps, and the total trajectory length is 69.

# G  Further Discussions

**Explanation of the performance gap with GWIL**    GWIL is a state-based method, but having access to expert states does not necessarily ensure superior performance. In the original GWIL paper [9], the authors provided simulation results on Pendulum-to-CartpoleSwingUp (Fig. 7 in [9]), showing GWIL achieves 600 points. However, Fig. 7 in [9] assumes that the target-domain learner receives the target-domain sparse reward directly in addition to the IL reward. So, the GWIL result in our paper can be different from that in the GWIL paper. Despite our efforts to tune GWIL's hyperparameters, GWIL did not show high performance only with the IL reward without direct target-domain sparse reward. Our approach consistently shows better performance in almost all tasks presented in this paper. As the GWIL paper indicates, GWIL can recover the optimal policy only up to isometric transformations. Given that expert demonstrations only cover a fraction of the optimal policy's scope, recovering the exact optimal policy becomes more difficult. Additionally, GWIL uses Euclidean distance as a metric within each space to compute Gromov-Wasserstein distance, which confines the method to scenarios limited to rigid transformations between domains.

**Broader Impacts**    The proposed learning model does not raise significant ethical concerns. However, as an IL algorithm, it should be applied to train models that learn tasks beneficial to society.

