# OpenReview forum: "Domain Adaptive Imitation Learning with Visual Observation"
_NeurIPS.cc/2023/Conference — NeurIPS 2023 poster_

### Official Review · Reviewer_V2Bv · 2023-06-30

**Soundness:** 3 good
**Presentation:** 3 good
**Contribution:** 2 fair
**Rating:** 5
**Confidence:** 3

**Summary:**

This paper aims to solve the domain-adaptive imitation learning problem, where the demonstrations are collected in a source domain, and the agent needs to learn policies in a target domain. Beyond that, the tasks are with high-dimensional visual observations. The authors have proposed a new framework based on dual feature extraction and image reconstruction to achieve behavior feature extraction and reward generation.

**Strengths:**

This paper aims to solve a very practical problem, since in real-world problems, we can hardly collect expert demonstrations in the target learning environment. Instead, collecting expert demo in a similar source domain is much easier.

**Weaknesses:**

The framework in Figure 3 is too much complex. The authors are encouraged to clarify their contributions in this paper and simplify the parts in the existing literature.

Previous works study the problem of domain adaptive imitation learning, and this work aims to solve the same problem with visual observations. The proposed approach augments the previous work with an encoder to deal with the high-dimensional visual observations. This idea seems a little straightforward.

**Questions:**

The experiment section is weak. The proposed approach is barely evaluated in limited simple tasks. If the authors could show that the proposed approach can work in more challenging domains, such as humanoid running, the quality of this paper could be greatly improved.

**Limitations:**

This paper has not discussed the limitations in the main body of this submission.

---

> ### Author Rebuttal · Authors · 2023-08-09
>
> Thank you for your valuable feedback.
>
> **1. Contribution of this paper**
>
> In summary, our contribution is that we propose a novel feature extraction model in the domain-adaptive IL with visual observations. We significantly improved the quality of behavior feature extraction by effectively eliminating domain-specific information from the behavior features. The salient components include dual feature extraction, dual cycle-consistency, and an externally located reward-generating discriminator for policy training. We will make this part more clear in the paper.
>
> We would like to emphasize that our approach is beyond  a simple combination of existing domain-adaptive IL with encoders. Our approach merges the merits of prior IL and image translation approaches uniquely. Our proposed method enhances behavior feature extraction with domain-specific information elimination and facilitates the transfer of expert knowledge from the source domain to the target domain. Furthermore, our method outperforms existing approaches across various scenarios. We demonstrated the efficacy of our work in qualitative and quantitative evaluations, including learning curves, ablation studies, and visualization of behavior features via t-SNE.
>
> **2. Experiment on more challenging domain mismatch**
>
> Addressing domain adaptive IL with visual observations remains a remarkable challenge. Learning from visual demonstrations is not an easy problem, especially when the adaptation between the source and target domain is required.
> Our method achieves superior performances to baselines across various types of domain mismatches. It is also important to note that our proposed method tackles more intricate situations than previous approaches. Sections E and F.3 explain the distinction of the setup for IP and RE tasks. Moreover, we examined our proposed method on IL tasks with a more challenging scenario, and the results strengthened the ability of scalability of our approach to more complex domains. Please refer to Common Response and Fig. 1 of the attached PDF.

---

> > ### Comment · Reviewer_V2Bv · 2023-08-17
> > **Response**
> >
> > Thank you for the response. I acknowledge that I have read the rebuttal.

---

> > > ### Author Response · Authors · 2023-08-20
> > > **Response to the Comment**
> > >
> > > Thank you again for your valuable time and effort to review our paper. We appreciate the reviewer's insights and suggestions to improve the quality of our work.

---

### Official Review · Reviewer_3tvh · 2023-07-04

**Soundness:** 3 good
**Presentation:** 3 good
**Contribution:** 3 good
**Rating:** 7
**Confidence:** 3

**Summary:**

Imitation learning from visual observations for domain adaptation. Given expert and non expert demonstrations in a source domain, and non expert demonstrations in a target domain, learn a reward and policy in the target domain via image consistency and domain and behavior feature consistency.

**Strengths:**

Domain adaptation in imitation learning remains an important problem. The paper tackles this challenging problem assuming access to visual observations but not to actions, other proxy tasks or expert demonstrations in the target environment. These assumptions allow for better generality. The problem statement, setup and method are clearly presented and the quality is high.

**Weaknesses:**

- The paper empirically demonstrates shifts in color, number of joints that can move, and action space. What are the general assumptions about the relation between the source and target domains? (There is some similarity, otherwise transfer would not be possible..)
- The papers uses visual observations, which is a more general setup than states. However, the proposed domains are quite simple (very few agents/objects in each, very simple backgrounds, etc.). How is the method expected to scale to more complex environments?

**Questions:**

In addition to the weaknesses above,

- How do the target policies compare to the source policies? E.g. in RE2 to RE3, does the policy learn to use all 3 links or does it apply a strategy with 2 links? Would suggest providing videos for qualitative analysis.
- In Figure 10(a), why is it desired that the source and target expert behavior features overlap? Shouldn't they require different features for different domains? (E.g. number of links.)
- Is there an explanation for the performance gap with GWIL (in Figures 5-9)? A state-based method should have more information about the environment.
- Please state the state/action space for every environment (for completeness and clarity).
- Suggest defining formally what good/bad actions mean.
- Line 22: suggest replacing 'should' with 'can'.
- Typos in lines: 55, 130, 194, 195, 355.

**Limitations:**

Please add to Section G the assumption re access to the target environment for training $\pi_{\theta}$ and in what cases will D3IL fail to learn a successful target policy.

---

> ### Author Rebuttal · Authors · 2023-08-09
>
> Thank you for your valuable feedback.
>
> **1. Assumptions about the relation between the source and target domains**
>
> It may be difficult to express the general assumption about the relation between domains clearly and mathematically. But, we think that the general assumption about the relation between domains will be that the source-domain and target-domain environments share a common objective, such as maintaining the pole upright position, reaching to the goal point, or running forward.
>
> **2. Experiment on more challenging domain mismatch**
>
> We anticipate that domain adaptive IL with visual observations will be helpful for addressing real-world scenarios, which include more complex situations. Application to real-world scenarios would be more challenging and intriguing, and we aim to address real-world scenarios in our future works. To assess the applicability of the proposed method in more challenging situations, we provide experimental results of experiments conducted under more challenging domain mismatch conditions. Please refer to the Common Response and Fig. 1 of the attached PDF for detailed explanations.
>
> **3. Qualitative Analysis on RE2 to RE3**
>
> In the target domain, the agent adapts to the robot’s unique structure and dynamics to perform tasks. For example, in the RE2 to RE3 task, the agent in the target domain learns to utilize all three links for effective task execution. As a qualitative analysis, we provide visual representations of a demonstration sequence in the source domain and a sequence generated by the learned policy trained in the target domain. Please see Common Response and Fig. 3 of the attached PDF for detailed explanations.
>
> **4. Why is it desired that the source and target expert behavior features overlap**
>
> In this paper, our aim is for behavior features to contain task-related information while excluding domain-specific information. For example, in the transition from RE2 to RE3, the number of links could be seen as domain-specific components. On the other hand, the goal position and correlation between the arm movements and the goal position can be treated as behavior-related components. Our objective is to achieve overlap between the behavior features of source and target experts, leading to domain-independent behavior features.
>
> **5. Explanation of the performance gap with GWIL**
>
> GWIL is a state-based method, but  having access to expert states does not necessarily ensure superior performance. In the original GWIL paper, the authors provided simulation results on Pendulum-to-Cartpole (Fig. 7 of their paper), showing GWIL achieves 600 points. However, Fig. 7 of their paper assumes that the target-domain learner receives target-domain sparse reward directly in addition to the reward from imitation learning. So, our GWIL result in our paper can be different from that in the GWIL paper. Despite our efforts to tune  GWIL's  hyperparameters, GWIL did not show high performance only with the reward from imitation learning without direct target-domain sparse reward.  Our approach consistently shows better performance in almost all tasks presented in this paper. As the GWIL paper indicates, GWIL can recover the optimal policy only up to isometric transformations. Given that expert demonstrations only cover a fraction of the optimal policy's scope, recovering the exact optimal policy becomes more difficult. Additionally, GWIL uses Euclidean distance as a metric within each space to compute Gromov-Wasserstein distance, which confines the method to scenarios limited to rigid transformations between domains.
>
> **6. State/action space for every environment**
>
> Here are the state/action dimensions for every environment. We will include this in the paper to improve clarity. We also include the size of observation for each environment. For the detailed explanation of visual observations, please refer to Appendix D.2 and E.
>
> - IP (state 4-dim, action 1-dim, observation 32x32 RGB)
> - IDP (state 11-dim, action 1-dim, observation 32x32 RGB)
> - RE2 (state 11-dim, action 2-dim, observation 48x48 RGB)
> - RE3 (state 14-dim, action 3-dim, observation 48x48 RGB)
> - HC (state 17-dim, action 6-dim, observation 64x64 RGB)
> - HC-LF (state 13-dim, action 4-dim, observation 64x64 RGB)
> - DMCS Cartpole (state 5-dim, action 1-dim, observation 32x32 RGB)
> - DMCS Pendulum (state 3-dim, action 1-dim, observation 32x32 RGB)
> - DMCS Acrobat (state 6-dim, action 1-dim, observation 32x32 RGB)
> - PointUMaze (state 7-dim, action 2-dim, observation 64x64 RGB)
> - AntUMaze (state 30-dim, action 8-dim, observation 64x64 RGB)
>
> **7. Meaning of good and bad actions**
>
> From the reinforcement learning perspective, good actions refer to ones that yield a larger return. From the imitation learning perspective, good actions refer to ones that closely resemble the expert actions in a given state. We will clarify this part in the paper.
>
> **8. Comments on Section G**
>
> As the reviewer pointed out, the proposed method requires interaction with the target environment for training. It is a characteristic, shared by several domain adaptive IL approaches. It will be interesting to extend our method to offline approach in future works. As mentioned in Section 2, some approaches do not require target-domain interactions during training, but those methods  rely on expert demonstrations for proxy tasks in the target domain.
>
> The success of the proposed method is affected by the similarity of the objective in the source and target domains. It will be difficult for our approach to succeed if source and target domains aim for completely different tasks. Future works could also include the extensions of our method to multi-task or multi-modal situations. We will include this in the paper.

---

> > ### Comment · Reviewer_3tvh · 2023-08-17
> > **Response to rebuttal**
> >
> > Thank you for the response, it has addressed and clarified some issues regarding the results. My score remains the same since it is still not clear how to generally characterize what is the required relationship between source and target domains to achieve generalization, or how well this method can scale to more real-world domains.

---

> > > ### Author Response · Authors · 2023-08-20
> > > **Response to the Comment**
> > >
> > > We sincerely thank you again for your valuable comments and suggestions. We appreciate the reviewers' time and effort to review our paper and help improve the quality of our manuscript.

---

### Official Review · Reviewer_szho · 2023-07-06

**Soundness:** 3 good
**Presentation:** 3 good
**Contribution:** 3 good
**Rating:** 6
**Confidence:** 4

**Summary:**

The paper attempts to solve the problem of domain shift in Imitation Learning with image observation. The author proposes D3IL architecture which is based on dual feature extraction (behavior and domain specific features from input) and dual cycle consistency (image level and feature level cycle consistency). The paper demonstrates effectiveness of the proposed method compared to baselines Third Person IL (TPIL), Disentan GAIL over different Gym based tasks (e.x. Reacher, Pendulum etc.) and  DeepMind Control Suite (DMCS) tasks (Maze based tasks).

**Strengths:**

- The problem addressed in an interesting problem for Imitation Learning.
- Good thorough evaluation different types of experiments.
- Strong experiments results.
- The paper is well written.


**Weaknesses:**

- The problem is well motivated from real robotics based task, but no real robotics based task is demonstrated in the paper. It would be interesting to see the application on sim-to-real based tasks.
- It would be good to discuss work similar TCN work [1].
- The extensions to other IL methods like IRL is a little unclear.
- Line 90-93 are a little unclear.

[1] Sermanet, Pierre, Corey Lynch, Yevgen Chebotar, Jasmine Hsu, Eric Jang, Stefan Schaal, Sergey Levine, and Google Brain. "Time-contrastive networks: Self-supervised learning from video." In 2018 IEEE international conference on robotics and automation (ICRA), pp. 1134-1141. IEEE, 2018.

**Questions:**

- Have you tried interchanging source and target domains?
- The relations to source -> expert and target -> learner is a little confusing. Maybe if you can represent in a better way that would be really good.



**Limitations:**

- most of them are mentioned in the paper

---

> ### Author Rebuttal · Authors · 2023-08-09
>
> Thank you for your valuable feedback.
>
> **1. Experiment on more challenging domain mismatch**
>
>  As the reviewer mentioned, our  method was evaluated primarily on simulated environments. During this short rebuttal period, we could not apply our method to real-world scenarios, but to assess the applicability of the proposed method to more challenging situations, we conducted more experiments with more challenging domain mismatch conditions. For the results, please see  the Common Response and Fig. 1 of the attached PDF therein. These results lead us to anticipate that our proposed method has enough possibilities to extend to more complicated domains including real-world scenarios.
>
> **2. Discussion of TCN**
>
> Time-Contrastive Networks (TCNs) learn representations from visual demonstrations of different viewpoints via time-contrastive approach. TCN makes embeddings from images of the same time but different viewpoint to be similar and makes embeddings from images of different times in the same viewpoint to be dissimilar. One of the key distinctions between TCN and our method is that while TCN relies on time-synchronized demonstrations with simultaneous viewpoints for model training and reward construction, our method does not necessitate any time-synchronization among demonstrations. Also, TCN mainly addresses viewpoint mismatch between domains, while our method encompasses general domain mismatch including the difference of viewpoint, degree-of-freedom, dynamics, and embodiment. We will include the discussion in the paper.
>
> **3. Discussion to the extensions to IRL**
>
> Our method involves an adversarial IL scheme designed for domain adaptive IL with visual observations. The feature extraction model learns to extract domain-independent behavior features from observations. Subsequently, the reward-generating discriminator $D_{rew}$ generates a reward by comparing the behavior feature of the policy to that of the expert demonstration.
> As most adversarial IL methods do, our approach also learns the reward for policy training. Therefore, we anticipate extending our approach to the IRL framework in future works, involving making a reward function itself capable of transfer across diverse domains.
>
> **4. Lines 90-93**
>
> In addition to expert demonstrations from the source domain, we assume access to the non-optimal trajectories of image observations from both the source and target domains. These non-optimal trajectories are generated by non-expert policies. The visual observations of rollouts from a non-expert policy $\pi_{SN}$ in the source domain are denoted by $O_{SN}$, and the visual observations of rollouts from a non-expert policy $\pi_{TN}$ in the target domain are denoted by $O_{TN}$. Here, 'SN' refers to 'source non-expert' and 'TN' refers to 'target non-expert'. We will clarify this part in the paper.
>
> **5. Interchanging source and target domains**
>
> As shown in the paper, we included experimental results where source and target domains were interchanged. The examples are IP-IDP, RE2-RE3 on OpenAI Gym (Fig. 5 in the main paper) and CartPole-Pendulum on DeepMind Control Suite (Fig. 7 in the main paper).
>
> **6. Relations to source-expert and target-learner**
>
> In this paper, the source domain is where we can obtain expert demonstrations, and the target domain is where we train the agent. The expert is the one who performs a task optimally. The agent in the target domain learns the task from the expert's demonstration  in the source domain. We call the agent or its policy a learner, since in the beginning it is not an expert for the target-domain task. We will clarify this part in the paper.

---

> ### Author Response · Authors · 2023-08-20
>
> Thank you again for your valuable feedback to review our paper.
> We kindly mention that the discussion period is about to end. We sincerely hope that our responses and additional experiments successfully address the reviewers' concerns and enhance the clarity of our work. If the reviewers have remaining comments and suggestions, we will be grateful to share them with us.

---

### Official Review · Reviewer_33qr · 2023-07-25

**Soundness:** 3 good
**Presentation:** 3 good
**Contribution:** 3 good
**Rating:** 5
**Confidence:** 4

**Summary:**

This work introduces a method of learning a visual behavior encoding for reinforcement learning tasks, D3IL.

From a TPIL behavior feature encoder baseline, D3IL introduces dual feature extraction (including both behavior and domain feature encoders) and dual cycle-consistency (cycle consistency on image and feature levels). From inputs of unpaired target and source images, D3IL translates across domains and then cycles back to the original domain, via a pair of generator networks which take in source images and domain targets. Further, image reconstruction consistency, feature vector similarity, and feature vector consistency is added for further loss and stability.

The domain and behavior feature encoders and target generator from above are taken and used to train a reward generating discriminator, $D_rew$, which distinguishes between expert and learner behavior (rather than expert and non-expert data used for training the previous models). This is used alongside the target behavior encoder to train SAC RL policies for the given tasks.

D3IL is tested on many domain transfer tasks with variations in visual effects, degree-of-freedom, internal dynamics, and robot embodiment - D3IL demonstrates superior performance on the target domain compared to baselines of TPIL, …. It is further demonstrated on an AntUMaze task where direct RL learning in the target domain is difficult, but possible with D3IL. Ablation studies show that the contributions of each of the consistency losses across feature and images, and cycle consistency, are all important and contribute to the final performance.


**Strengths:**

#### Originality
The work builds on top of TPIL to build a larger framework for cross-domain imitation / feature encoding, by incorporating dual feature extraction and cycle consistency, and training a further reward-generating discriminator.

#### Quality
The experiments are detailed, covering four categories of domain change and with ablation studies on components of the loss. The experiments are run over multiple seeds and the method is compared against several baselines (TPIL, DisetanGAIL, GWIL) to present a good picture of the overall learning and domain transfer capability. There are further many more detailed experiments in the appendix to which present good coverage.

#### Clarity
The work is generally well-written and the description of D3IL which draws from premises of prior TPIL work is understandable. The explanations of losses and model composition are detailed and the diagrams help with understanding the proposed methodology. The experiments are easy to follow. Further task explanation and experimental methodology is well explained in the appendix.


#### Significance
The method seems broadly applicable across tasks and domain transfer goals in simulation.

The experiment on AntUMaze is particularly motivating as towards the practical applications of the method, demonstrating that a task which could not be solved directly could be improved by training from an imitation agent learned on a simpler task - in a sort of curriculum learning.


**Weaknesses:**

#### Quality
The dual feature extraction, in particular the domain encoder added from TPIL, is motivated on lines 135-136 as to “check that the obtained feature well contains the required information without information loss, based on image reconstruction and consistency check.” However, it’s still unclear why specifically it is necessary to pass observations as input to domain encoder, as the work assumes that the policy knows what environment it is operating in. Would an ideal domain encoder be able to output a one-hot vector indicating its domain to minimize all the domain-encoder relevant losses? As there are separate generators for each domain (source/target), should it be clear without any domain conditioning what the “style” of each generator output should be? For example, in CycleGAN and its successors, the domain encoding is not required.

An ablation study without using a domain encoder, but a static method of encoding the domain or skipping the domain encoding entirely may be helpful.

#### Clarity
Unfortunately many of the experimental results are left to to the appendix including all of Sections 6.2 and 6.5, and reading the appendix seems necessary to have a full understanding of the model performance.

The method is somewhat hard to initially grasp from reading the text of Section 5 as there are many similar variables in the notation which differ by one letter or modifier, with many such modifiers. However, explanations are generally helped by Figures 1 and 3.

The limitations are also only mentioned in the appendix.

Misc:
- Line 191: is another form of consistency [that] guide[s] the model so that …
- Line 195: scheme significantly improve[s]
- Line 211: we need [the] expert behavior feature

####  Significance
The practical applications of the method seem best suited for cases where it is hard to train an expert in the target environment directly via RL, but having access to expert data on other tasks as observation-only would help. Many of these cases would be covered by real world scenarios, which is out-of-scope of the current work. Even on the AntUMaze problem, the learning process did not necessarily have to be done via adversarial learning and visual-only observations, as the full expert state would be available.

From the ablation in Table 1, it appears that the large majority of the performance improvement comes from image-level consistency and the feature-level consistency was only of minor gain, especially compared to the uncertainty/error.

As mentioned in the limitations in the appendix, the paper also introduces quite the complex process with significantly many more loss terms and two pipeline stages compared to prior work, which may make it harder to apply. Adversarial learning techniques generally require more careful tuning than supervised methods, and this work introduces two adversarially-learned stages.


**Questions:**

Some discussion on the weakness-quality section would be most helpful


**Limitations:**

Some consideration of the limitations is explained in the weaknesses-significance section.

---

> ### Author Rebuttal · Authors · 2023-08-09
>
> Thank you for your valuable feedback.
>
> **1. Answer to Quality section**
>
> We claim that learned domain encoders are essential for enhancing the feature extraction and policy performance in the domain-adaptive IL problem with visual observations. To demonstrate this,  we conducted an experiment wherein we excluded domain encoders and their corresponding discriminators from our proposed model while keeping all other elements consistent. We then trained the policy in the target domain to evaluate its performance. The result is shown in Fig. 2 of the attached PDF of the Common Response. As depicted in the figures, our proposed model with domain encoders is superior to our model without them. To enhance domain-independent behavior feature extraction, we proposed employing both domain encoder and behavior encoder, ensuring that the resultant domain feature and behavior feature are mutually independent and contain complementary information about the input. This approach was motivated by the observation that solely relying on a behavior encoder was insufficient to eliminate domain-related information from the behavior feature.
>
> **2. Answer to Clarity section**
>
> We made efforts to incorporate as much content as possible within the main body of the submission. Nevertheless, due to the extensive volume of the contents and the constraints of the main body, some details and results had to be left in the appendix. However, we ensure that the most crucial contents and results are present in the main paper. We will mention the discussion of limitations in the main body.
>
> **3. Experiment on more challenging domain mismatch**
>
> We anticipate that domain-adaptive IL with visual observations will be helpful in real-world scenarios. As the reviewer mentioned, our  method was evaluated primarily on simulated environments. During this short rebuttal period, we could not apply our method to real-world scenarios, but to assess the applicability of the proposed method to more challenging situations, we conducted more experiments with more challenging domain mismatch conditions. For the results, please see  the Common Response and Fig. 1 of the attached PDF therein. These results lead us to anticipate that our proposed method has enough possibilities to extend to more complicated domains. While we presented experimental results on the simulated environments due to time constraints, we aim to address real-world scenarios in our future works.
>
> **4. Feature-level consistency**
>
> As the reviewer  pointed out, the impact of image-level consistency on the performance of the learned policy appears to be more prominent than that of feature-level cycle-consistency. Nevertheless, it's important to emphasize that both image-level cycle-consistency and feature-level cycle-consistency are essential in extracting domain-independent behavior features. It is also worth noting that the policy achieves the best performance when all these elements are combined, and including feature-level cycle-consistency doesn't incur additional complexity to the model structure.
>
> **5. Complexity**
>
> As discussed in the paper, our proposed framework involves a two-stage learning process that can increase complexity. Moreover, similarly to our proposed method, adversarial learning generally requires careful handling to ensure stability. However, the complexity of the proposed approach is much less than  that of widely-used image translation models because our model's network structure consists of only a few layers of convolutional neural networks and multi-layer perceptrons. Additionally, we observed that effective loss component balancing and discriminator regularization with gradient penalties are sufficient for achieving favorable results.

---

> ### Author Response · Authors · 2023-08-20
>
> Thank you again for your valuable feedback to review our paper.
> We kindly mention that the discussion period is about to end. We sincerely hope that our responses and additional experiments successfully address the reviewers' concerns and enhance the clarity of our work. If the reviewers have remaining comments and suggestions, we will be grateful to share them with us.

---

### Author Rebuttal · Authors · 2023-08-09

Thank you for your valuable feedback. In the Common Response, we address inquiries from multiple reviewers and provide responses requiring images and plots. Please refer to the attached PDF for detailed explanations.

**Experiment on more challenging domain mismatch**

To assess the effectiveness of our proposed method in more challenging scenarios, we tested our method on IL tasks with domain shifts caused by variations in robot morphology. We employed Walker, Cheetah, and Hopper environments in the DeepMind Control Suite. In these environments, the objective is to propel a 2D robot forward as fast as possible by manipulating its joints. The difficulty arises from the significant difference in the robot morphology and dynamics between the source and target domains.

We evaluated our proposed method and baseline algorithm in three tasks: Walker-to-Cheetah, Walker-to-Hopper, and Cheetah-to-Hopper (denoted as 'A-to-B' where 'A' is the source domain and 'B' is the target domain). A camera generates image observations by observing the robot's  movement. Rewards were assigned to the agent based on the robot's horizontal velocity, with higher velocity leading to greater rewards. Fig. 1 of the attached PDF depicts the learning curve of our method and baselines on three tasks, where each curve shows the average horizontal velocity of the robot in the target domain over three runs. These results demonstrate the superiority of our method over the baselines. The results in the paper and the attached PDF lead us to anticipate that our proposed method can extend to more complicated domains. While we presented experimental results on the simulated environments due to time constraints, we aim to encompass real-world scenarios in future works.

As highlighted in the paper, domain adaptive IL with visual observations is still a challenging problem. Each task considered  in the Experiments section exhibits a distinct domain mismatch, and our proposed method significantly enhanced performances on various tasks without necessitating time-aligned data or proxy task demonstrations. Consequently, our work plays a substantial role in advancing the field of domain adaptive IL with visual observations.

**Ablation study on domain encoders (Reviewer 33qr)**

During the rebuttal period, we conducted an experiment wherein we trained our proposed feature extraction model of D3IL without involving domain encoders ($DE_S$ and $DE_T$) and their corresponding discriminators ($DD_D$ and $BD_D$). All other networks and losses remain unchanged. We then trained the policy in the target domain to assess its performance. We evaluate our method with and without domain encoders in four tasks: IP-to-two, RE2-to-three, RE3-to-tilted, and CartPoleBalance-to-Pendulum. Fig. 2 of the attached PDF shows the results. The results demonstrate the  benefit of conditioning generators on learned domain encoders to boost model performance. To enhance domain-independent behavior feature extraction, we proposed employing both domain encoder and behavior encoder, ensuring that the resultant domain feature and behavior feature are mutually independent and contain complementary information about the input. This approach was motivated by the observation that solely relying on a behavior encoder was insufficient to eliminate domain-related information from the behavior feature.

**Qualitative Analysis on RE2 to RE3 (Reviewer 3tvh)**

In the target domain, the agent adapts to the robot's unique structure and dynamics to perform tasks. For example, in the RE2 to RE3 task, the agent in the target domain learns to utilize all three links for effective task execution. As a qualitative analysis, we provide visual representations of a demonstration sequence in the source domain and a sequence generated by the learned policy trained in the target domain. In Fig. 3 of the attached PDF, we present sample trajectories of image observations. The red mark in the figure indicates the goal position. The top row corresponds to a demonstration in the source domain (RE2), while the bottom row corresponds to the policy learned in the target domain (RE3) over 1 million timesteps. A similar analysis for the UMaze environment is available in Appendix F.7.

---

### Decision · Program_Chairs · 2023-09-21

**Decision:**

Accept (poster)

**Comment:**

Imitation learning from visual observations is an important topic within the NeurIPS community, and the authors have provided strong empirical evidence that the proposed method represents an important new entry in this space. That said, the authors are strongly encouraged to make the modifications to the paper that arose during the author-reviewer discussion paper, especially one that makes it clear that the questions from reviewers regarding how to measure/quantify domain shift remain unanswered.